# Quantifying disorder one atom at a time using an interpretable graph neural network paradigm

James Chapman [1,2] ✉, Tim Hsu [3] ✉, Xiao Chen[3], Tae Wook Heo [2] & Brandon C. Wood[2] ✉

Quantifying the level of atomic disorder within materials is critical to understanding how evolving local structural environments dictate performance and durability. Here, we leverage graph neural networks to define a physically interpretable metric for local disorder, called SODAS. This metric encodes the diversity of the local atomic configurations as a continuous spectrum between the solid and liquid phases, quantified against a distribution of thermal perturbations. We apply this methodology to four prototypical examples with varying levels of disorder: (1) grain boundaries, (2) solid-liquid interfaces, (3) polycrystalline microstructures, and (4) tensile failure/fracture. We also compare SODAS to several commonly used methods. Using elemental aluminum as a case study, we show how our paradigm can track the spatio-temporal evolution of interfaces, incorporating a mathematically defined description of the spatial boundary between order and disorder. We further show how to extract physics-preserved gradients from our continuous disorder fields, which may be used to understand and predict materials performance and failure. Overall, our framework provides a simple and generalizable pathway to quantify the relationship between complex local atomic structure and coarse-grained materials phenomena.

Understanding how a material's structure affects its properties is one of the most fundamental principles in materials science. At the center of this paradigm is the fact that macroscopic material's behavior begins at the atomic scale, with local atomic arrangements ultimately coming together to form structural features observed at larger length scales[1,2]. Characterizing the nature and propagation of these local environments is therefore vital to understanding macroscale structure-property relationships and their evolution[3]. Complicating this endeavor is the fact that the long-range features often depend on structurally disordered atomic environments, which tend to dictate materials functionality[4]. For instance, transport, chemical reactivity, and phase nucleation are all profoundly affected by the presence of interfaces, interphases, and grain boundaries[5–10]. These processes, in turn, are intricately connected to performance-durability trade-offs in both functional[11] and structural[12] materials. Examples include temperature-dependent microstructure evolution[13,14], hotspot formation[15,16], and the nucleation and growth of new material phases[17,18].

However, quantifying local atomic disorder in a physically motivated way in practice is extraordinarily difficult[19,20]. Although a number of methods have been proposed to characterize local atomic environments, these methods are often not optimized to magnify the subtle differences present in disordered environments. Existing methods can typically be grouped into three general classes, each of which carries distinct trade-offs: (1) semi-empirical structure factors such as Adaptive Common Neighbor Analysis (CNA)[21], Steinhardt order

[1]Department of Mechanical Engineering, Boston University, Boston, MA, USA. [2]Materials Science Division, Lawrence Livermore National Laboratory, Livermore, CA, USA. [3]Center for Applied Scientific Computing, Lawrence Livermore National Laboratory, Livermore, CA, USA. ✉e-mail: jc112358@bu.edu; hsu16@llnl.gov; wood37@llnl.gov

parameters[22], Ackland-Jones order parameters (AJ)[23], atomic excess volume[24], Centrosymmetry parameters (CSP)[25], Scalar Graph Order Parameter[26], and the local atomic environment metric[27]; (2) parameterized symmetry functions such as the Smooth Overlap of Atomic Positions[28], Behler-Parinnello functions[29], Moment Tensor Representations[30], Polyhedral Template Matching (PTM)[31], Distortion Factors[32], and the Adaptive Generalizable Neighborhood Informed functions[33]; and (3) unsupervised machine learning methods which include graph-based[34–37], order parameter-based[38,39] and image-based[40,41] representations.

In general, it is highly desirable to develop a methodology that is, by construction, specifically designed to distinguish, quantify, and physically interpret regions with varying degrees of atomic disorder. Such a capability would enable more accurate predictions of how disordered atomic environments translate to higher-level features and functionality. For instance, mapping between discretized atomistic models and continuous field representations, such as phase-field[42–44] and finite-element[45] models, forces the use of ill-defined and arbitrary approximations[46], particularly when the disorder is present. Moreover, continuous field representations propagate via local gradients[47], the evaluation of which amplifies inaccuracies associated with disordered regions. Addressing these shortcomings is, therefore, a critical priority.

To this end, we introduce a physics-aware workflow composed of two stages, which can be seen in Fig. 1. First, we use graph neural networks (GNN) to explicitly encode local atomic structural information. Next, we apply this encoding to map the local atomic structure to an order parameter that characterizes the local disorder. This order parameter, henceforth referred to as the Structural Orderness Degree for Atomic Systems (SODAS), $\lambda_i$, quantifies an atom's local structure in terms of the "closeness" to likely environments encountered between two limiting cases: a perfect crystal ($\lambda_i = 1$) and a melt ($\lambda_i = 0$). Our approach offers three distinct advantages: (1) the graph representation accurately encodes the topology of the connected network of atoms; (2) our paradigm is universally tunable to specific material systems that exhibit temperature-dependent structural transitions; and (3) the physical interpretability of atomic-level predictions due to the bounding of the problem between physically-identifiable endpoints. The power of this workflow is demonstrated by application to several examples of disordered aluminum systems, including solid-solid and solid-liquid interfaces, polycrystalline microstructures, and fracture evolution, and is compared to other methods from the literature such as CNA, AJ, PTM, and CSP.

## Results
### Definition of SODAS

In principle, for systems that exhibit temperature-dependent structural transformations between phases A and B, the level of the configurational disorder can be mapped onto an equivalent level of thermal disorder in a finite-temperature ensemble. To this end, we can introduce a fictitious temperature ($T'$) that mathematically represents this configurational disorder. In practice, $T'$ can be parameterized for a given system using explicit MD simulations, as discussed in the Methods section. To physically bound $T'$, we introduce $T_d$ as the limit of full disorder (nominally the melting temperature). The value of $T'$ is then confined to the range between 0 and $T_d$. We next define $\gamma$ as a global structural order parameter:

$$\gamma(T'; T_d, s) = \mathcal{N} \frac{1}{1 + \exp(-(T_d/T')^s)} \tag{1}$$

where $\mathcal{N}$ normalizes $\gamma$ between 0 (absolute disorder) and 1 (absolute order), defined as $\mathcal{N} = \sigma_\gamma(max_\gamma - min_\gamma) + min_\gamma$. $s$ is an empirical scaling metric that determines where to begin the decay of $\gamma$ from ordered to disordered. The introduction of $s$ makes the definition of $\gamma$ universal when considering systems that exhibit temperature-dependent structural transformations, as one can simply tailor its value for any unique material system. One can think of $s$ as a way to control the steepness in the drop-off between order and disorder for a specific material system. For this work, $s$ was set to 1.5. It is important to note that $\gamma$ is defined as the global level of disorder for an entire material system and not at the atomic scale. A further discussion of the relationship between $\gamma$ and $s$ can be found in the supplemental information.

While $\gamma$ describes the level of disorder of a macroscopic, homogeneously disordered system, we are primarily interested in local atomic disorder within a heterogeneous system. To establish a connection between the global and local scales, we map the likelihood of finding a given local atomic environment within an ensemble of configurations to a local order parameter (SODAS), $\lambda(n)$, where $n$ indexes an atom. One can think of $\lambda(n)$ as defining which temperatures you are most likely to find a given atomic environment, but mapped to a point along a phase space trajectory between two phases, A and B. In practice, due to ergodic constraints, we assume that this ensemble can be sampled discretely from MD simulations. We represent this

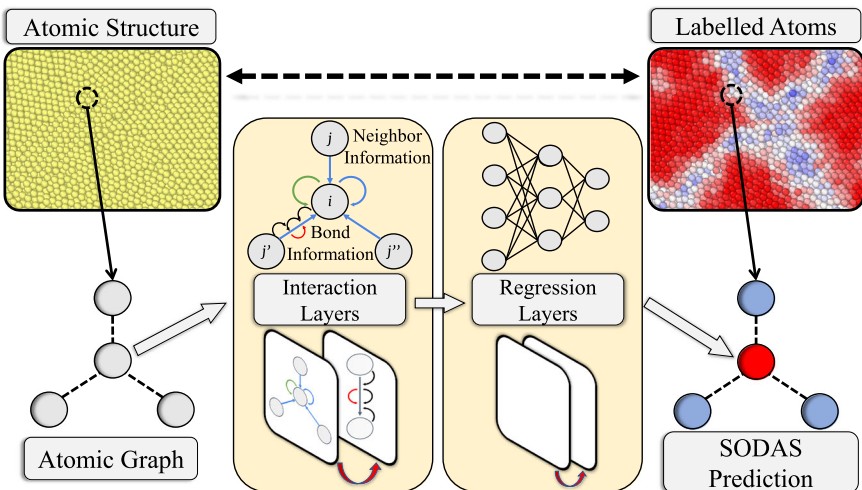

**Fig. 1 | General workflow for calculating SODAS values.** Atomic structures are converted into graph representations, which explicitly encode all necessary geometric information. These atomic graphs are then fed into a graph neural network, which has been trained to distinguish between the unique local geometries in different material phases. The graph neural network then gives each atomic environment a SODAS value, which classifies where in the phase space, between phases, that local structure is most likely to occur.

mapping as:

$$f(\{\gamma\}_i) \mapsto \lambda(n) \qquad (2)$$

where $\{\gamma\}_i = \{\gamma_1, \gamma_2, \ldots, \gamma_k\}$ represents the set of $\gamma$ values associated with a given local atomic structural motif $i$ across a discrete set of $k$ ensembles, and $f$ is a function that maps $\{\gamma\}_i$ to $\lambda(n)$. It is important to understand that we are arguing a local atomic environment may be represented as the set of points along the order-to-disorder spectrum, rather than its geometric symmetries (or lack thereof). We would argue that this definition provides a more grounded representation of the local environment as it can be mapped back onto a physical system, rather than an unsupervised feature vector. While the function $f$ is unknown, it can be approximated. In this work, we use a graph neural network scheme to facilitate this approximation, while retaining physical interpretability. It is also important to understand that the GNN is not predicting the temperature of a local environment but rather the point along the order-to-disorder spectrum that the local environment is most likely to exist at. One may think of this scheme as optimizing a high-dimensional non-linear function that maps the temperature of the system to the point along a path between two phases in a representative configuration space. We also note that $s$ can be iteratively tuned for a given system by using an initial guess and observing the error between the predicted average $\lambda$ for a structure and the theoretical gamma at the known thermostat temperature during training. Figure 1 outlines the key steps in this process and is discussed in further detail in the methods section.

## Validation of SODAS

We first validate the SODAS model by observing where the average SODAS value within several bulk configurations at different temperatures align with the theoretical values of $\gamma$, as seen in Fig. 2. Here, we see that all atoms in the structure at 0 K is uniformly predicted to have $\lambda = 1$, which is indicative of the perfect crystal. In contrast, at 1200 K, all atoms indicate $\lambda$ to be close to 0 due to the structure existing as a melt. On average, structures between these limits yield intermediate values of $\lambda$, as expected. In all cases, the average value of $\lambda$ aligns well with the theoretical value of $\gamma$, providing evidence that our methodology, both in conception and implementation, is accurate. A red horizontal line is also drawn in Fig. 2 to provide the reader with a clear understanding of where absolute disorder is represented along the y-axis.

At the same time, detailed visualization of intermediate-temperature configurations reveals a spectrum of atomic environments covering a range of $\lambda(n)$ in lieu of homogeneously distributed

disorder. For example, the second structure in Fig. 2, which represents a structure at roughly 200 K, has $\lambda$ values ranging from 0.9 to 0.99. Accordingly, as described previously, similar atomic environments exist at a range of temperatures but with different degrees of expression according to the average overall level of disorder. Intuitively this makes sense, as the goal of the SODAS metric is to judge the likelihood of an atomic environment existing at an arbitrary point along the abstract spectrum between fully ordered and fully disordered variants. If a unique atomic environment occurs at multiple temperatures, one would expect its $\lambda$ to be a weighted combination of the individual occurrences of the environment along the temperature spectrum.

## Boundary identification in solid-solid interfaces

While the perturbed pristine bulk structures provide a case study for us to analyze how SODAS performs, most interesting structures contain defects, and varying levels of disorder. To this end, in this section, we analyze how SODAS can be used to extract structural information out of solid-solid interface regions. Since $\lambda$ is continuously valued over the discrete atoms, it can be interpolated to a continuous field. This mapping allows for its integration into continuum models. For instance, we note the similarity between such a continuous field representation and the phase order parameter used in phase-field models[8,48]. We showcase this concept for the example of two grain boundary regions with varying levels of interfacial complexity. Nevertheless, we note that this method can be used for other classes of crystalline interfaces, such as symmetric tilt and twin boundaries, and edge/screw dislocations.

From Fig. 3, one can see the intuitive nature of SODAS, cleanly characterizing the grain regions with a $\lambda$ close to 1, smoothly transitioning to higher degrees of disorder present near the boundary. For boundaries that show higher degrees of crystallinity, such as those in $\Sigma5(110)[120]$, the disorder present at the interface is minimal, as is expected, though is still clearly present. Likewise, for more disordered boundaries, such as those in $\Sigma9(110)[110]$, a greater degree of disorder is detected within the interface region. As in the previous section, these characterizations exemplify the ability of SODAS to determine where the grain begins and ends.

Figure 3 shows the continuous fields derived from the originally discrete, per-particle SODAS value $\lambda$. Additionally, the gradient norm $\|\nabla\lambda\|$ was calculated and visualized. This discrete-to-continuum conversion was done by interpolating the discrete $\lambda$ values onto a uniform grid using PyVista[49]. When calculating the gradient of this field, we observe areas of the structure where there are sharp changes in the SODAS values. Notably, the gradient is maximized not at the center of the grain boundary, but rather at the transition to the boundary region, because these are locations within the structure where there is an abrupt change in the level of disorder present.

The sensitivity of this detection can be seen in Fig. 3, where the gradient of the scalar field predicts two regions where there is an abrupt change in the SODAS values. As we move from the crystalline regions towards the interface normal to the boundary region, we first encounter a crystal-to-boundary region, followed by the boundary itself, and finally, a boundary-to-crystal region as we move away from the interface. Therefore, the gradient predictions in Fig. 3 highlight the fact that a degree of homogeneity can exist in both the ordered interior of the grain as well as the disordered interior of the grain boundary.

## Boundary identification in solid-liquid interfaces

Disordered interface boundaries are notoriously difficult to quantitatively characterize due to the inherent complexity and heterogeneity in their local atomic environments. Bond-angle methods such as CSP, CNA, AJ, and PTM sometimes struggle to accurately distinguish between perturbed crystalline and disordered atomic environments[50,51]. These difficulties ultimately make defining interface

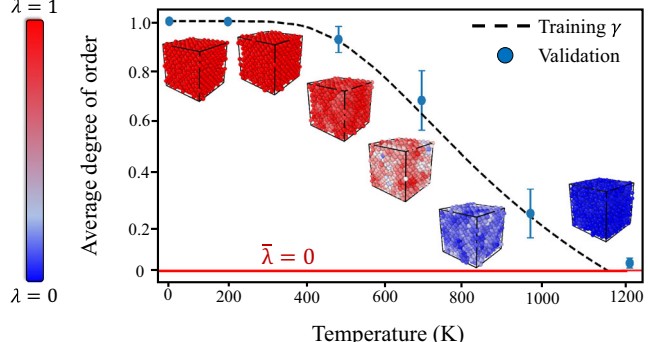

**Fig. 2 | SODAS calculations on bulk structures taken during a superheating MD simulation.** Values along the y-axis represent the average SODAS value for each shown structure, whose atoms are colored according to each atom's SODAS value. The dashed line indicates the theoretical values of $\gamma$ while the plotted SODAS values represent the accuracy of the GNN mapping. The red horizontal line at $y = 0$ indicates a point of absolute disorder. The error bars represent the standard deviation of $\lambda$ values predicted at a given temperature.

boundaries challenging. In contrast, the SODAS formalism accomplishes this by providing a continuous metric that allows for a physically justifiable and mathematically rigorous definition of the interface boundary transition.

To this end, we have performed two-phase crystal/liquid CMD simulations at several temperatures (100, 500, and 1200 K), to observe

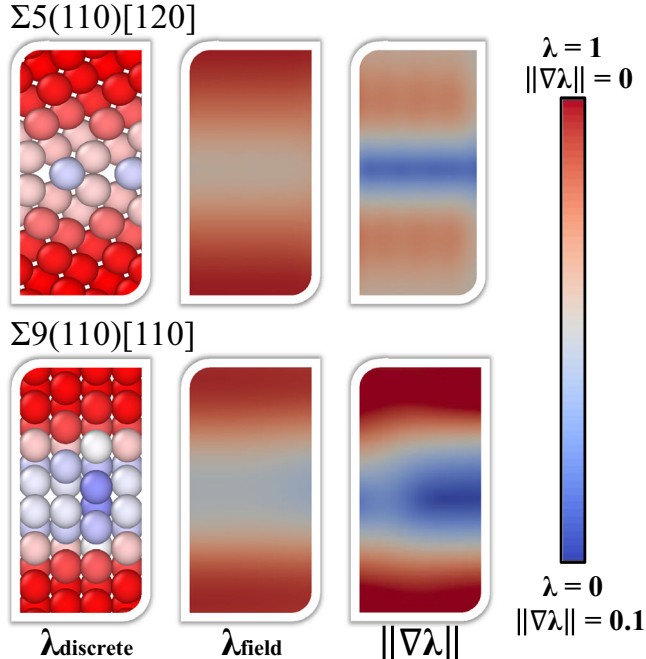

**Fig. 3 | SODAS predictions and gradients of grain boundaries.** Continuous fields (and its gradient norm) of the originally discrete, per-particle SODAS value $\lambda$. The discrete-to-continuum conversion is done by interpolating the discrete $\lambda$ onto a uniform, fine grid. The gradient information can then be computed over the uniform grid.

how several methods classify the unique structural environments present in each scenario. Further details regarding the simulation setup can be found in the Methods section. Previous works have shown that the solid-liquid boundary in Al exhibits a soft transformation when moving from the solid to the liquid, with a gradual increase in the level of disorder as a function of distance from the solid phase[52,53]. These results indicate that a three to four atomic-layer boundary exists between the solid and liquid phases in which the level of disorder continuously increases as one approaches the liquid phase. This implies that one needs a characterization method that can continuously and smoothly map the topology of the interface boundary.

Figure 4 provides a comparison between SODAS, PTM, and CSP. Here, we examine both the total structure, which again represents the solid-to-liquid transition, as well as a zoom-in view of the interface boundary itself. Figure 4a shows the SODAS characterization of the solid-to-liquid transition, along with a zoomed-in interface boundary region. Here, SODAS correctly identifies the crystalline region as corresponding to structure typically found at low temperatures, as this region is at 100 K in Fig. 4a. The interface boundary region in the zoomed-in portion also shows a natural gradual progression from solid-to-liquid, which one would expect at equilibrium conditions. There are regions where the interface is more crystalline, and regions that are more disordered, with a gradual degradation between these regions. This highlights that SODAS can accurately quantify both the solid and liquid phases, as well as the interface boundary between them.

Figure 4b reference the predictions made by CSP. Here, while CSP does an excellent job of identifying the crystalline region, it's characterization of the liquid phase seems less reliable due to the misclassification of various sites throughout the liquid region. This misclassification stems from the fact that CSP works on the notion that values close to zero represent highly ordered crystal structures, while values away from zero represent deviations from those crystal symmetries. While CSP clearly indicates all atoms in the liquid as being away from the corresponding crystal symmetry, it fails to truly quantify liquid environments from one another. Beyond some threshold, a CSP far away from zero does imply that it is more structurally dissimilar than a value closer to zero, but again beyond some threshold. Therefore, it cannot distinguish liquid environments from one another. From Fig. 4b, one can also see a more discrete characterization of the

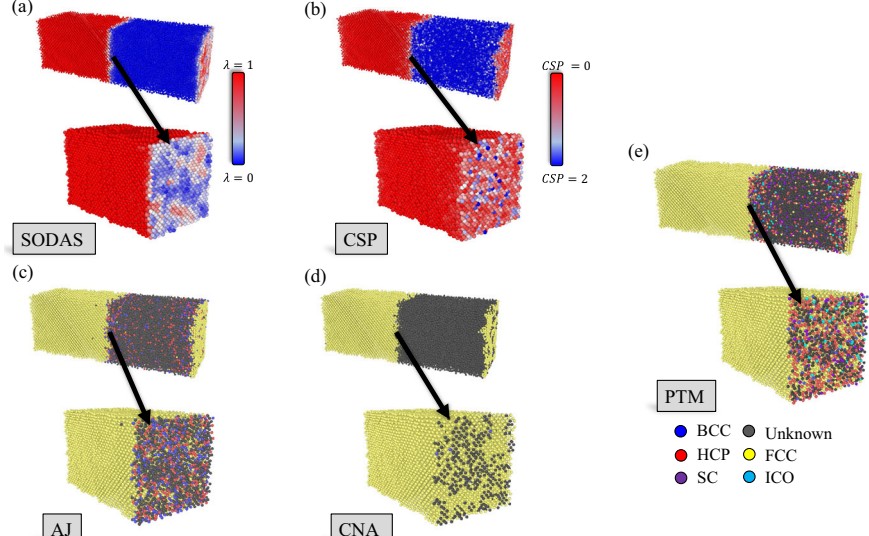

**Fig. 4 | Characterization of solid-liquid interfaces.** Comparison between SODAS (**a**), CSP (**b**), AJ (**c**), CNA (**d**), and PTM (**e**) when quantifying the interphase interface region between a solid and a liquid. Each subplot provides a full view of the SODAS characterization of the solid-liquid interface with a zoomed and sliced view of the interface boundary transition when going from the solid phase to the liquid. SODAS and CSP have their own colorbars, shown within the subplot, while (**c**–**e**) all share the same labels, which are shown in (**e**).

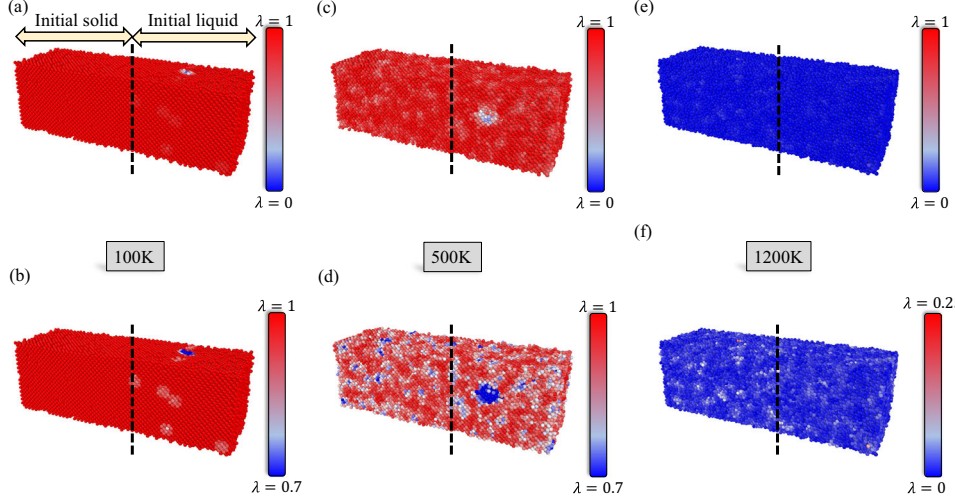

**Fig. 5 | Characterization of equilibrated solid-liquid interfaces at different temperatures. a** SODAS characterization on the final structure of the solid-liquid interface at 100 K. Inserted labels indicates which side of the structure was initially a solid and which side was initially a liquid. These inserts apply to all sub-figures in this figure. **b** The same characterization as (**a**) but with a different SODAS color bar scale. **c** SODAS characterization on the final structure of the solid-liquid interface at 500 K. **d** The same characterization as (**c**) but with a different SODAS colorbar scale. **e** SODAS characterization on the final structure of the solid-liquid interface at 1200 K. **f** The same characterization as (**e**) but with a different SODAS colorbar scale. The atoms in each system are color-coded based on their SODAS values, as referenced in their respective colorbars. Note that not all colobars have the same scale.

interface boundary region, where only a vague guess can be made regarding regions of the boundary that are more liquid-like versus solid-like. As these regions would differ greatly in energy, and, therefore, properties, one can reason that CSP is not capable of providing an accurate description of this region.

For the cases shown in Fig. 4c–e, which covers AJ, CNA, and PTM, respectively, we examine how the binary classification schemes perform when characterizing the solid-liquid interface region. In all cases, the solid region is well defined, though, in both AJ and PTM, the liquid region disordered atoms are often classified as a particular solid phase, indicating a breakdown in the classification algorithm. Within the interface region, both AJ and PTM give a seemingly random characterization of structure types. While CNA performs better with a clear mapping between the SODAS and CNA classifications present within the interface, CNA provides a more coarse level of information, giving only the impression of solid-like and liquid-like regions. Overall, SODAS provides a significantly finer and more informative prediction of the solid-liquid interface region.

While Fig. 4 qualitatively compares the continuous nature of the solid-liquid interface, Fig. S5 provides a more quantitative picture. Here, we examine the average order parameter value, normalized between 0 and 1, for all methods as a function of distance along the y-axis, which exhibits the solid-to-liquid transformation. Figure S5's highlights are as follows: (1) CSP provides a reasonable deviation in order parameter values when moving away from the solid, but fails to quantify the end of the boundary interface and liquid phase, (2) PTM and AJ both indicate the existence of a one to two atomic-layer thick boundary interface, short of the experimentally observed length[52], and also provide a jagged set of values towards the ends of the boundary interface region, and (3) while a-CNA and SODAS both predict a boundary interface region in the experimentally observed range, only SODAS provides a smooth gradient throughout the entire process, confirming our qualitative analysis in the paragraphs above.

Figure 5 shows the SODAS characterization of the atomic environments present in the final MD configurations for the three temperatures described earlier. Figure 5a shows the SODAS predictions on the 100 K system, with inserted arrows indicating which regions of the structure were initially crystalline and which regions were initially

liquid. One can observe, on the initial liquid side, the presence of several disordered regions. These regions are more clearly shown in Fig. 5b due to the specified SODAS range in the color bar. Intuitively it makes sense that the portion of the structure that was initially a liquid would have defects upon quenching to 100 K, while the region that was initially crystalline would not have such defects at 100 K.

Figure 5c shows the SODAS predictions on the 500 K system. The inserted arrows from (a) are not shown here but are implied, with the black dashed line from (a) being present. One can observe, on the initial liquid side, the presence of many disordered regions, with one large disordered patch in the middle. These regions are more clearly shown in Fig. 5d due to the specified SODAS range in the color bar. Again, this makes sense that the portion of the structure that was initially a liquid would have defects upon quenching to 500 K. Importantly, SODAS identifies larger defects present in the initial liquid region than in the initial crystal region, which also makes sense as local environments that lead to larger defects are easier to access kinetically in the liquid phase than they are in the crystal phase. While there are perturbed regions in the initial crystal portion, there are no large-scale defects present.

Figure 5e shows the SODAS predictions on the 1200 K system. The inserted arrows from (a) are not shown here but are implied, with the black dashed line from (a) being present. As 1200 K is above this interatomic potential melting temperature, SODAS correctly identifies the total structure as being in the liquid phase. However, we observe patches in the structure that are less disordered than others. These regions are more clearly shown in Fig. 5f due to the specified SODAS range in the colorbar. Interestingly, from Fig. 5f, one can see that the regions exhibiting less disorder are more common in the regions that was initially a solid, which makes sense as some level of structural similarity with the solid phase could be present upon melting. It is also possible that the initial crystal region has not yet reached equilibrium with the initial liquid phase. In either case, SODAS captures this trend and allows for a pathway for more complex analysis.

**Autonomous microstructural feature extraction**
The ability to define boundary transitions also enables the identification of larger-scale microstructural features. To demonstrate this

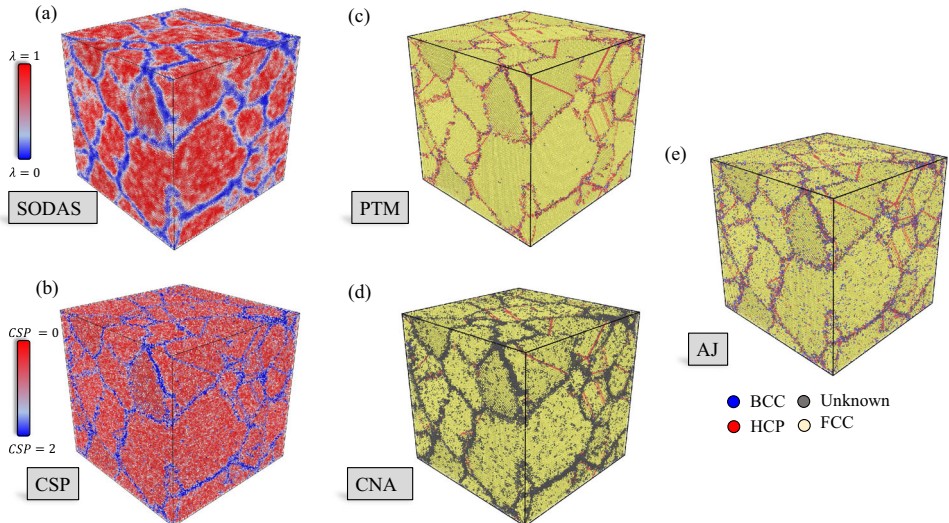

**Fig. 6 | Final snapshot along the 600 K CMD polycrystalline trajectory for several methods. a** SODAS, **b** CSP, **c** PTM, **d** CNA, and **e** AJ. SODAS colorbar shown below (**a**), the CSP colorbar shown above (**b**), and (**c–e**) labels shown below (**e**).

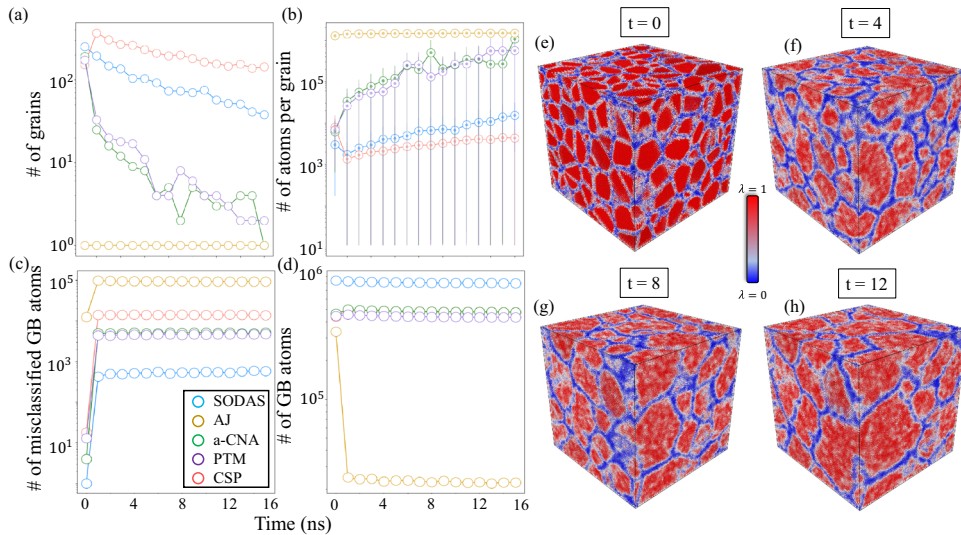

**Fig. 7 | Polycrystalline structure evolution during CMD at 600K. a** Number of grains as a function of time. **b** Number of atoms per grain as a function of time, where error bars represent the largest grain (top bar) and smallest grain (lower bar). **c** Number of misclassified grain boundary atoms throughout the structure as a function of time. **d** Number of grain boundary atoms as a function of time.

**e–h** SODAS Characterizations of several polycrystalline structures at various times throughout the CMD simulation. The SODAS colorbar is shown in the middle of the four structures, with each structure labeled according to the time the snapshot was taken along the CMD trajectory.

capability, we showcase the performance of SODAS when compared to CSP, a-CNA, AJ, and PTM, for the classification of both grain boundary and grain regions in dynamic polycrystalline structures. As described in the Methods section, we performed an MD simulation at 600 K of a 1.6 million atom FCC aluminum system containing 250 initial grains.

Figure 6 provides a visual comparison of the final MD snapshot between the various characterization methods. Here, many grains have coalesced to form larger grains over time as enough kinetic energy was present in the system to overcome large potential energy barriers. Figure 6a provides the SODAS visualization, where one can clearly identify the grain boundary regions (shown in blue), the grains (shown in red), and the transition region between the grain and grain boundary (shown in white). Since we are defining grain boundary atoms as atoms having a certain level of disorder, this visualization provides a simple thresholding procedure, as one can clearly identify regions

based on their level of disorder/order. This prescription will result in some misclassified atoms, however, as the temperature is increased due to the level of disorder present within the grain regions. The level of misclassification can be quantified, however, and is shown in Fig. 7.

Figure 6b–e provide the visualizations of CSP, PTM, CNA, and AJ, respectively. CSP (b) has difficulty identifying the transition region between grain and grain boundary due to the level of noise present within its characterization. Here, many atoms are misclassified as grain boundary atoms, and it is clear that there is no obvious threshold that would allow for precise identification of the two regions. PTM (c) performs better within the grain than CSP; however, it performs worse within the grain boundary region. This is due to the inherent level of disconnectedness found within the boundary regions, where there are significant levels of noise when attempting to discern which atoms belong to the grain boundary. This should be

no surprise, given PTM's struggles to classify the boundary region of the solid-liquid interface.

CNA, shown in Fig. 6d, provides a better depiction of the GB regions than PTM, though it has difficulty within the grains. CNA yields deviations when identifying structure types as a level of atomic perturbations, with larger perturbations leading to larger errors in its characterizations of structure types. Therefore, while the boundaries themselves are reasonable, the transition between boundary and grain is not smooth and continuous. Finally, AJ (e) provides extremely thin and disorganized boundary regions with chaotic misclassification of grains present throughout the structure.

Figure 7 aims to quantify how both the grain boundaries and grains evolve over time. Figure 7a depicts how the number of grains changes as a function of time for all methods considered in this work. Here, we can see that SODAS was the only method to correctly identify all 250 grains in the initial configuration. As the temperature is increased, SODAS predicts a gradual decrease in the number of grains, which coincides with a gradual increase in the number of atoms within the grain regions, as shown in (b). The number of atoms within the grain boundary regions should also decrease in a similar manner, as atoms are leaving the grain boundaries and moving into the grain regions. This is evidenced in (d), where SODAS predicts a similar decrease in the number of grain boundary atoms over time as the growth of grain atoms. (c) shows the number of misclassified atoms, as described in the Methods section. From (c), we can see that SODAS has the smallest number of misclassified atoms present in the system for all cases except the initial configuration. This is because we've chosen a $\lambda = 0.7$ as our threshold, but at $t = 0$, all atoms in the grain region presumably have a $\lambda = 1$. Due to this, there are additional atoms in the transition region between grain and grain boundary that will get grouped with grain boundary due to their smaller $\lambda$ value, but actually belong in the grain. While this leads to a larger number of misclassified atoms in the initial structure, it is important to note that this number is consistent with the remaining structures as the system evolves.

For CSP, Fig. 7a indicates a smaller number of grains captured in the initial configuration, followed by an increase in the number of grains to more than the original number. This is due to CSP's noisy characterization, in which many small grains, on the order of a few tens of atoms, are being classified as their own grain instead of belonging to a single larger grain. This leads to a smaller average grain size, as shown in (b). CSP does capture the gentle decrease in the number of grains over time, though. Accordingly, CSP shows a much larger number of grain boundary atoms present in the system, as shown in (d), which we expect due to the noisy characterization. CSP also leads to a larger misclassification number (c), which aligns with and explains our noisy characterization argument.

For both CNA and PTM, Fig. 7a shows a smaller number of initial grains followed by a drastic dropoff as time evolves. This is due to the lack of a true thresholding parameter in the binary classification scheme these methods employ, yielding a more rigid definition of grain and grain boundary atom. While the number of atoms per grain increases in (b), it increases much more sharply than either CSP or SODAS. One can also see a much larger number of misclassified atoms in (c), again due to issues characterizing perturbed local structures at non-zero temperatures. There is also an increase in the number of grain boundary atoms when compared to SODAS, though the reduction in the grain boundary atoms over time does follow a similar trend.

Finally, for the case of AJ, Fig. 7a shows only a single grain present throughout the entire simulation, including the initial structure. This is due to AJ suffering even worse thresholding issues that either CNA or PTM, leading to an unphysical blending of the grain boundary and grain regions. This is also seen in (b), with there being a large number of atoms per grain, as AJ only finds a single grain in the system. This trend aligns with the number of grain boundary atoms in (d), with AJ

registering nearly an order of magnitude fewer grain boundary atoms than any other method. There is also a larger number of misclassified atoms shown in (c), leading to a larger amount of blending between the two domains.

## Dynamic fracture evolution

Here we examine the performance of various methods at capturing the initiation of tensile fracture. We quantify performance as the ability of a given method to accurately predict the location of shear bands throughout the material using only the level of disorder captured by each method. We use $D^2$, which has been used previously to gauge shear band locations[54], as a way of judging the accuracy of the structural characterization methods used in this work. $D^2$ is effectively capturing the instantaneous measure of an atom's local displacement, making it a decent proxy for local irreversible shear transformations. Figure 8 shows the results of this quantification on snapshots throughout the MD simulation. Figure 8a–d provides insight into the location of shear bands by observing the locations in the structure, along the z-axis, that represent the highest levels of disorder for a given method. For all methods, kernel density estimation (KDE)[55] is used to determine peaks in each method's disorder predictions. In the case of $D^2$, CSP and SODAS, values fed into KDE correspond to predicted values at a given z-coordinate in the top 5% of disorder. For a-CNA, PTM, and AJ, all values not characterized as a known crystal structure are used as their characterization scheme is binary. Figure 8e–h visualize the snapshots in (a–d) using SODAS to color-code the atoms. Figure S4 provides a visualization of other methods over the course of the MD simulation.

Figure 8a shows the approximated predicted shear band locations (peaks in the $D^2$ distribution), for several methods at 1 ns into the MD simulation. Several interesting points can be made here: (1) CSP provides a flat distribution, implying that it detects a uniform level of local structural change, (2) PTM only predicts a single peak at small z-coordinates while a-CNA and AJ predict the location of three bands in the bottom half of the structure, and (4) neither SODAS nor $D^2$ predicts the formation of any bands in this structure. These results seem to indicate that CSP, a-CNA, PTM, and AJ are extremely sensitive to local structural changes when compared to SODAS and $D^2$. Figure 8b shows several major changes, including the appearance of shear bands according to $D^2$, with SODAS peaks aligning reasonably well with peaks predicted by $D^2$. (b) also shows that all other methods produce a nearly flat distribution, again showing the difficulty in predicting shear band locations with these methods due to their seemingly random prediction of disorder along the z-axis. Interestingly, both AJ and a-CNA in (a) are in decent agreement with $D^2$ in (b), though their predictions fall out of disagreement in (b), indicating that these two methods could be used to predict future locations of band formations but not instantaneous locations of them. It is also important to note that in both (a) and (b), the structure has not undergone tensile failure, and represents the lead-up to the eventual failure of the material.

Figure 8c highlights an instance after the material has fractured, though not into two separate pieces yet. The semi-transparent yellow region in (c) indicates the location of fracture initiation. Here, we see a significant shift in the band locations and corresponding densities, which makes intuitive sense given the extreme structural changes in the material. (c) highlights the agreement between $D^2$ and SODAS over a nearly 225 Å range of the z-axis. All other methods show a nearly uniform prediction of three band peaks, which disagrees with the $D^2$ prediction of two peaks. The disagreement occurs between 75 and 150 Å, which represents the regions in the material where the fracture is occurring. This would indicate that all other methods deviate from $D^2$ in regions where the level of disorder is extreme, perhaps due to the mischaracterization of such environments. We note for clarity that SODAS and $D^2$ remain in excellent agreement throughout the entire z-axis.

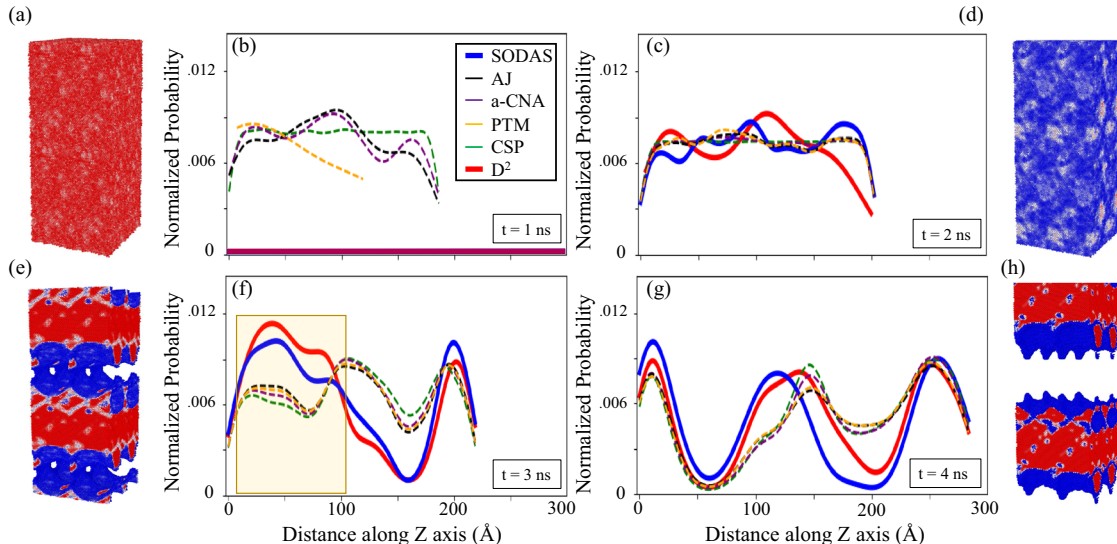

**Fig. 8 | Characterization of dynamic fracture evolution simulations.**
**a**–**d** provide insight into the location of the shear band by observing where each method predicts the highest levels of structural disorder in the system, compared to a ground-truth value of $D^2$. Colors in (**a**–**d**) represent the different methods, with inserted boxes showing the time during the simulation that the snapshot was taken from. **e**–**h** provide visual depictions of the fracture process, with colors representing the SODAS values at each snapshot. Each image represents the atomic structure captured by the shear band location plots. The semi-transparent yellow region in (**c**) indicates the location of fracture initiation.

Figure 8d shows the shear band locations after the material has fractured into two pieces. Here, we see reasonably good agreement between all methods, though we do note that SODAS still provides the most accurate picture when compared to $D^2$. We note again, however, that AJ, CSP, PTM, and a-CNA predict the approximate band locations given by $D^2$ in (d) in the previous panel, (c), indicating that these methods show promise in predicting future trends in potential band locations, but do not necessarily serve as accurate instantaneous band predictors.

To summarize these findings, we note three main takeaways: (1) All methods outside of SODAS predict the approximate formation of shear bands before their actual formation, according to $D^2$, indicating their potential use as future predictors of shear behavior but not instantaneous ones, (2) during the fracture process SODAS provides the most accurate depiction of potential shear band locations when compared to $D^2$, and (3) due to the alignment of SODAS and $D^2$ over the majority of the MD simulation, we conclude that the deviation from $D^2$ for all methods outside of SODAS is likely due to misclassification of highly disordered regions in the material, implying that one must be capable of accurately capturing those regions to truly understand the mechanical properties during the failure process.

## Discussion

In summary, characterizing the nature of the local atomic disorder is critical and necessary to understand how structure-property relationships evolve. SODAS is a new mathematical framework in which local atomic environments are transformed into graph representations, encoded via a graph neural network paradigm, and finally mapped onto a local order parameter. This order parameter, $\lambda$, is an informative, continuous, and mathematically bounded scalar which represents the level of disorder present within an atomic environment, and is analogous to an atomically resolved configurational entropy density. In addition to the examples shown throughout this work, these advantages allow for the universal quantification of a multitude of complex and heterogeneous materials properties and phenomena.

We also envision our proposed methodology as a tool for multiscale model integration. In particular, SODAS provides an atomistically derived, physically motivated continuous scalar field representation for phase field and continuum models. This mapping can be likewise leveraged to output field quantities such as phase order, grain distribution, concentration, stress/strain, and so on. Such an approach offers a new perspective and valuable technique for bridging scales in multiscale models, both between atomistic and microscale descriptions, as well as between discrete and continuous representations. We further emphasize that although this work focuses on single-element systems, our method is generally applicable to multi-component systems and their corresponding microstructural features.

The advantages of SODAS also become clear for extraction of physical properties that relate to materials' performance or degradation. For instance, we showed that by interpolating the discrete representation to a continuum representation, we could analyze or differentiate $\lambda$ to deduce spatially resolved changes in structural homogeneity. In practice, these structural changes often map to changes in key response properties, including diffusivity, dielectric response, electrical conductivity, and elastic compliance[56]. In cases where such properties can be computed locally or measured using local probes, SODAS offers a way to extract analytical relationships between structure and function. Moreover, sharp gradients from abrupt changes in response functions can concentrate electrical, chemical, or mechanical potential, creating hotspots that can initiate key electrochemomechanical failure modes. We, therefore, propose that gradients in the continuous representation of $\lambda$ may provide a robust way to identify such hotspots, with a direct connection to early prediction of the propensity for deleterious outcomes such as fracture, corrosion, and thermal runaway.

## Methods
### Training data preparation
Classical molecular dynamics (CMD), using the LAMMPS software package[57], was used to generate training data for the GNN model. Starting from bulk FCC aluminum (containing 1024 atoms), CMD was performed in the NVT ensemble using Zhou et al. EAM potential[58]. The range of temperatures used for the training data was 50 to 1200 K. At each temperature, an NVT simulation was performed for 10 ns. Data used for training was taken after the 5 ns mark to ensure that only equilibrated configurations were used for training.

## Graph neural network implementation

**Conversion to graph.** Prior to GNN operation, we converted the atomic systems into graphs using a simple cutoff radius-based neighbor list search (implemented using Atomic Simulation Environment[59]), with the cutoff $R_c = 3.5$ Å. Each node of the converted graph corresponds to the atom type $z$, and each edge the bond distance $d$. In the end, our graph representations encode the atoms and their local neighbors, with atoms represented as nodes, and neighbor connections represented as edges between nodes.

**GNN operation.** The GNN model used in this work consists of three components: the initial embedding, the atom-bond interactions, and the final output layers (Fig. 1). In the initial embedding, each atom type $z$ is transformed into a feature vector by an `Embedding`, layer (PyTorch[60]). Each bond distance $d$ is expanded into a $D$-dimensional feature vector by the Radial Bessel basis functions (RBF)[61],

$$\text{RBF}_n(d) = \sqrt{\frac{2}{R_c}} \frac{\sin(\frac{n\pi}{R_c}d)}{d}, \qquad (3)$$

where $n \in [1..D]$ and $R_c$ is the cutoff value. Both atom and bond feature vectors have the same length $D = 100$.

The atom-bond interactions are also known as GNN convolution, aggregation, or message-passing. There are many variants of GNN convolution operations that can be adopted from the literature. In this work, we choose the edge-gated graph convolution[62,63]. The term atom-bond interaction is based on the fact that the nodes and the edges exchange information during the convolution operation. Specifically, the node features $\overrightarrow{h}_i^{l+1}$ of node $i$ at the $(l+1)$th layer is updated as

$$\overrightarrow{h}_i^{l+1} = \overrightarrow{h}_i^l + \text{SiLU}\left(\text{LayerNorm}\left(\overrightarrow{W}_s^l \overrightarrow{h}_i^l + \sum_{j \in \mathcal{N}(i)} \hat{\overrightarrow{e}}_{ij}^l \odot \overrightarrow{W}_d^l \overrightarrow{h}_j^l\right)\right), \quad (4)$$

where SiLU is the sigmoid linear unit activation function[64]; LayerNorm is the layer normalization operation[65]; $\overrightarrow{W}_s$ and $\overrightarrow{W}_d$ are weight matrices; the index $j$ denotes the neighbor node of node $i$; $\hat{\overrightarrow{e}}_{ij}$ is the edge gate vector for the edge from node $i$ to node $j$; and $\odot$ denotes element-wise multiplication. The edge gate $\hat{\overrightarrow{e}}_{ij}^l$ at the $l$th layer is defined as

$$\hat{\overrightarrow{e}}_{ij}^l = \frac{\sigma(\overrightarrow{e}_{ij}^l)}{\sum_{j' \in \mathcal{N}(i)} \sigma(\overrightarrow{e}_{ij'}^l) + \epsilon}, \qquad (5)$$

where $\sigma$ is the sigmoid function, $\overrightarrow{e}_{ij}^l$ is the original edge feature, and $\epsilon$ is a small constant for numerical stability. The edge features $\overrightarrow{e}_{ij}^l$ is updated by

$$\overrightarrow{e}_{ij}^{l+1} = \overrightarrow{e}_{ij}^l + \text{SiLU}\left(\text{LayerNorm}\left(\overrightarrow{W}_g^l \overrightarrow{z}_{ij}^l\right)\right), \qquad (6)$$

where $\overrightarrow{W}_g$ is a weight matrix, and $\overrightarrow{z}_{ij}$ is the concatenated vector from the node features $\overrightarrow{h}_i$, $\overrightarrow{h}_j$, and the edge features $\overrightarrow{e}_{ij}$:

$$\overrightarrow{z}_{ij} = \overrightarrow{h}_i \oplus \overrightarrow{h}_j \oplus \overrightarrow{e}_{ij}. \qquad (7)$$

Lastly, via the final output layers, each node feature is eventually transformed into a scalar output $y$ ranging from 0 to 1. In this work, these final output layers are a two-layer multilayer perceptron (MLP) with SiLU activation after the first layer ($D = 100$ neurons) and sigmoid output after the second layer (scalar output). Effectively, the GNN

predicts the SODAS metric for every atom. Further details regarding model training are described in Supporting Information.

## Atomistic simulation details

All CMD simulations were performed using the LAMMPS software package[57] and the Zhou et al. EAM potential[58].

**Two-phase simulations.** Two-phase simulations were performed within the NVT ensemble by creating two initial, independent thermostats within a rectangle block of Al containing roughly 51,000 atoms. Both regions are initially crystalline. During the initial stages of the MD simulations, the independent thermostats are used to create a liquid region and a crystal region. Within the liquid region, the thermostat is set to 2000 K, which the crystal region is set to 100 K. After 5 ns, allowing for equilibration of the independent phase regions, the two thermostats are removed and replaced by a single new thermostat which acts on the entire system. This thermostat is set to different temperatures depending on the different scenarios considered. These temperatures are 100, 500, and 1200 K. The combined system is equilibrated using the single thermostat for 5 ns.

**Grain coarsening simulations.** CMD simulations in the NVT ensemble were performed for four polycrystalline cases, each with a varying number of initial grains. An initial bulk aluminum system containing roughly 1.6 million atoms was used to construct a polycrystalline system, using the Atomsk software package[66], containing 250 initial grains. CMD simulations were performed at 600 K. NVT simulations were run for ~1.5 ns for each combination of initial structure and temperature. Further details regarding the polycrystalline structures can be found in the Results section as well as the Supplemental Information.

## Microstructure characterization

Microstructure characterization occurs in four stages: (1) calculation of SODAS for all atoms in the system, (2) thresholding of the atomic configuration based on an atom's SODAS value and subsequent removal of all atoms below the threshold value, (3) conversion of the remaining atoms to a graph representation for the discovery of subgraphs within the graph. Analysis of these subgraphs, such as the calculation of the number of atoms in the subgraph, were then performed. Figure S3b depicts this workflow visually. While step (1) requires little-to-no input from the user, step (2) requires one to define the level of disorder that needs to be captured when defining the interface regions. As the threshold value defines the structural properties of the interface region itself, with a near-zero threshold indicating grain boundaries which are extremely disordered and a value close to 1 representing highly crystalline boundaries. In principle, both classes of interfaces can exist within the same structure, which would require a more complex thresholding system, though for this work, we assume a uniform local atomic environment amongst all grain boundaries.

For all microstructure characterization tests in this work, we employ a thresholding technique to differentiate between atoms belonging to a grain and atoms belonging to a grain boundary. For the case of SODAS, we define this threshold as 0.7, as this value of $\lambda$ corresponds roughly to the level of disorder one would expect at 600 K. The threshold values at each temperature are defined in a way that minimizes the number of atoms within a grain that may be mistaken as grain boundary atoms. At 600 K, which is roughly half of the melting temperature, there is a significant amount of kinetic energy in the system, which causes non-trivial levels of atomic perturbation.

The same thresholding technique is used for SODAS is also used for CSP. Like SODAS, CSP works on the assumption that non-zero values of CSP represent deviations from a symmetric known crystal structure. However, it is not clear whether or not this trend holds the

further one moves from a CSP of zero, implying no obvious cutoff value to distinguish order from disorder. Similar to the SODAS case, we set the 0K CSP threshold at $10^{-5}$, and all other CSP thresholds are set at 2, meaning anything CSP value greater than 2 will be considered a grain boundary atom. For PTM, CNA, and AJ, the atomic-level characterization is different than that of CSP and SODAS. Here, environments are encoded as a one-hot vector of known crystal phases. If the atomic environment does not belong to a known reference, it is classified as "unknown". Here, any atom classified as "unknown" is determined to be within a grain boundary.

Once atoms in the system have been thresholded, and all grain boundary atoms have been removed, the remaining system is then mapped onto a graph, $G$, shown in Fig. S3, where edges are represented by ij pairwise interactions within a 4 Å cutoff radius. A recursive subgraph search algorithm is employed to discover all connected subgraphs, $S_G$, within the complete graph $G$. This algorithm is extremely efficient, discovering all subgraphs within a 1.6 million atom system in 1.2 s. As all interface atoms were removed prior to the graph construction, all subgraphs in $G$ represent the resulting grains contained within the structure. The total number of subgraphs in the system is equal to the number of grains in the system, and the number of nodes in each subgraph is the number of atoms in a given grain.

We also examine the grain boundary atoms and identify the number of such atoms that have been misclassified. We define this miss-classification as atoms belonging to the grain boundary designation that do not have a required number of neighbor connections within a cutoff of 3 Å. Recall that here we are only examining grain boundary atoms. Atoms within the true grain boundary should have a large number of grain boundary neighbors, whereas atoms that have been classified as grain boundary atoms but actually lie somewhere within a grain should have a fewer number of grain boundary atoms as neighbors. We use the combination of the number of grains detected, atoms per grain distributions, and miss-classified grain boundary atoms to judge a given method's accuracy.

**Tensile fracture simulations.** A rectangular block of 64,000 atoms was given tensile strain at a constant strain rate of $0.5 \frac{\%}{ps}$, at 100 K under NVT conditions. The tensile strain was given in the z-direction, and the simulation box was allowed to change along that axis, while the box was held constant along the remaining two axis. The simulation was run for 5 ns. Calculations of $D^2$ were done using the OVITO software package[67].

**Visualization.** All atomistic visualizations were created using the OVITO software package[67]. Atoms-to-continuum visualizations were done using PyVista[49].

### Characterization method details

For PTM, a root mean square deviation of 0.25 was used. For CNA, an adaptive cutoff radius was employed. For CSP, the number of neighbors was set to 12 since we examined only the FCC phase of Al. We employed the minimum-weight matching convention for CSP. A CSP cutoff of 2 was used as the boundary between order and disorder, due to the location of the first CSP distribution peak.

## Data availability

Due to the file sizes of the data in this work, all data required to reproduce these results can be requested by contacting the corresponding author.

## Code availability

The SODAS code can be downloaded at https://github.com/LLNL/graphite. A detailed demo of model training and testing can be found at https://github.com/LLNL/graphite/blob/main/notebooks/sodas/training-and-inference.ipynb.

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

## Acknowledgements

J. Chapman, T. Hsu, X. Chen, T. W. Heo, and B. C. Wood are partially supported by the Laboratory Directed Research and Development (LDRD) program (20-SI-004) at Lawrence Livermore National Laboratory. This work was performed under the auspices of the US Department of Energy by Lawrence Livermore National Laboratory

under contract No. DE-AC52-07NA27344. J. Chapman also acknowledges the support of the Department of Mechanical Engineering at Boston University.

## Author contributions

X.C. and B.W. supervised the research. J.C. performed all MD simulations and devised/implemented the autonomous microstructure feature extraction methodology. T.H. trained the GNN and performed all GNN-related predictions. J.C. and T.H. devised the theoretical SODAS framework. B.W., T.H., and J.C. devised the atoms-to-field mapping, while T.H. implemented it. T.W.H. provided insight into the connection between atomistic and phase-field modeling and helped guide discussions surrounding the atoms to continuous field methodology. J.C. and T.H. wrote the manuscript with inputs from all authors.

## Competing interests

The authors declare no competing interests.
