## [Peer Review File · Nature Communications]

Quantifying Disorder One Atom at a Time Using an Interpretable Graph Neural Network ParadigmREVIEWER COMMENTS

Reviewer #1 (Remarks to the Author):

NCOMMS-22-22606: Quantifying Disorder One Atom at a Time Using an Interpretable Graph Neural Network Paradigm

This paper examines a new structural metric called SODAS and uses it to identify the effective temperature of atoms in a solid.

I write a strong review here with many criticisms. I wish to clarify that I think the metric and even more so, the use of graph neural networks, is a clever and innovative idea that could be useful for machine learning. I think there is a lot of potential in this work. However, as written, it falls short of expectations for Nature Communications. Key among the criticisms is that insufficient information is provided to make the work repeatable; I strongly encourage the authors to make code available or at least have a statement on code availability. Second, many claims about how the method is superior to other methods are given without providing evidence to justify those statements; many are noted in the comments below. In particular, if the authors want to show how much better it is, they could actually calculate the other metrics that are supposedly inferior so they can be contrasted.

The literature review is sufficient but I do note one recent paper that gives a bit more perspective on the diverse nature of structural metrics that have been developed (<https://doi.org/10.1021/acs.chemrev.1c00021>); the authors are encouraged to consider how they might redefine their three categories based on this work. There's also some work that considers deviations from solid structures with a local distortion factor that the authors are encouraged to consider how their work is unique since it is also a continuum value (doi: 10.1038/s41467-020-15619-9). Also, other work has used GNNs before too (<https://www.nature.com/articles/s41524-021-00650-1>).

Bottom of page 2 (Would be great to include line numbers next time), "facilitating universal and intuitive applicability across all atomic structures of a material". Note that this statement could be deceiving because it can only consider two limiting structures, though perhaps the authors are stating that it can contrast between the two limiting structures regardless of the structure being analyzed. However, in a material that exhibits solid solid phase transformations, would one of these be considered to be more like a liquid than another? Consider these thoughts and how that statement might be revised to accurately portray the potential of the method.

End of first paragraph of 2.2, "On average, structures between these limits yield intermediate values of λ , as expected." I find this statement a bit misleading since 1200 K clearly is not as expected. The authors note this later, but they have supplemental Fig 2 that could have easily been incorporated in Figure 2 that would indicate the challenge of SODAS metric in approaching the "liquid" structural arrangements.

Nothing is stated anywhere about whether the simulations have been equilibrated and how that

equilibration process occurs. This is particularly relevant for the Solid-Liquid interface simulations. The authors note in the methods (a detail that should be noted in results text) that the liquid region is at 4000K. How do the authors maintain pressure constant when the two parts are at such different temperatures? Does that lead to negative pressures in the solid region and positive pressures in the liquid region? How does the equilibration (temp and pressure) look when the shared thermostat is then employed? On the one hand it shouldn't be surprising that at 200 K all solidifies or at 1500 K all liquifies, but how was equilibrium obtained prior to these changes? While these don't affect the SODAS analysis, these are odd simulations and it is not clear how typical equilibrium conditions in these and the other simulations have been achieved.

First paragraph of 2.3, "Bond- angle methods such as CNA and AJ, and more complete methods such as SOAP, often fail to distinguish between perturbed crystalline and disordered atomic environments." The authors offer absolutely no evidence, either by reference or data in their paper, to support this statement. This should be removed unless the authors can provide supporting evidence. Furthermore, while the authors show they can identify the boundary regions with this method, they have not noted anything that shows it to be superior to the other methods mentioned.

I have some difficulties with any discussion noting that this method can identify environments. Certainly it is capable of doing that, but because of how it is trained, (from 4.1), "During training, all atoms at a given temperature are assigned the same value of γ . In this way, the value of γ is not directly connected to an atom's local structural geometry. However, at each temperature local atomic geometries bounce around an equilibrium point, which in this case can be thought of as the OK structure. As the temperature increases the magnitude of the displacement from this structure increases, though some atomic neighborhoods may resemble low temperature structural motifs, even at higher overall temperatures." The training is based on temperature, not structures. So, when an atom is assigned a SODAS value, it is its relevance to temperature or common structures at that temperature. The authors note that it could have that same structure at another temperature so the SODAS value will be different. Furthermore, lots of environments are all being assigned the same temperature. Don't get me wrong, I like the idea of this descriptor, but I think the authors need to be clear about what can actually be distinguished and what can't. What it is trained to capture is temperature, via the structures at each temperature, but thus it is finding lots of structures with the same temperature (and it does this well), but that means the SODAS can tell that the structures of a given temperature exist, not distinguish between different structures, like CNA, SOAP, etc. can do.

Colorbar in Figures 3 and 4 is all red, doesn't go red to blue as suggested it should by the atoms. Also the references in the text to Figure 5a and b are reversed and should be corrected. Did the figure 6 colorbar lambda values get switched? In all the other graphs, lambda=1 was the crystalline region.

While the authors have included many of the necessary details for someone to repeat their work, important details are missing that would make the work very hard to repeat. First off is the missing description for equilibration of the MD simulations. Second, the authors suggest multiple regression layers in Figure 1 but don't ever say exactly how many. I don't think the authors are trying to be

intentionally vague, but there is a lot of detail missing or it is hard to follow since it is an involved process. I think the authors must either describe the exact process in more detail, perhaps in supplemental if necessary, or they must provide a script as supplemental material so others can reproduce even a small part of their work. Both are recommended since if they provide a script, people are more likely to apply their methods in the first place. Additionally, it should be clarified that because of feature sizes, the trained model for SODAS calculations can only be applied to systems with the same number of atoms, right? Another example of missing details is the conversion to an atomic graph mentioned in the second paragraph of 2.4; while the method is published elsewhere, it is not clear how that atomic graph is different from the GNN of this work. Furthermore, the reference paper [38] is also missing lots of details so the reference doesn't appear to be sufficient for reproducibility.

Second paragraph of 2.4, "This process is extremely challenging using existing methods due to their lack of a continuously and bounded metric." This is another unfounded statement, at least with no reference given or evidence shown in the paper to support it. I will also note that there is work in this area that has been incorporated into OVITO for grain segmentation (<https://doi.org/10.1088/0965-0393/24/5/055007>), so while the SODAS method is impressive, it's not the only tool out there.

Third paragraph of 2.4, "As a result, the distribution of SGOP values provides a physically-intuitive and robust measure of the microstructure. This featurization is a more accurate and unique representation compared to the number of atoms in a grain, the grain radius, or the grain density." This is another unfounded statement. Grain size and grain radius are both very physically intuitive and I am not even sure what the value of SGOP means. Perhaps it is a robust measure of microstructure but it's not clear to me what it is a measure of whereas I know exactly what grain radius is.

Fourth paragraph of 2.4, "[Figure 5 shows] the interplay between temperature and grain size." The plot does not show grain size, it shows θ_G , which I can only assume is SGOP. SGOP may be a surrogate for grain size, but it is not grain size.

Fifth paragraph of 2.4, "However, a plateau is reached within 2 ns, and no further increase in the mean can be seen." Figure 5a $\log(\mu)$ has noise that is about the order of any increase or decrease. So, the authors either need to recognize this noise and say any changes are within this noise, or the changes or not noise and each change up or down has meaning. You can't have both and therefore the quoted statement needs to be adjusted for accurate description of the data. This goes for other comments about data in these graphs of Figure 5.

Figure 5 has a product of mean and standard deviation to "gauge both the rate of grain growth and the magnitude of the grains themselves." Is this an accurate statement? Is this a common statistical value used to capture those kinds of changes? Or is this a creation of the authors and if so, how do they suggest that mean multiplied by standard deviation should capture both rate of growth and the magnitude of the grains themselves since it has units of SGOP^2 ?

Figure 5b grain coarsening of the 5 grain structure. It seems obvious that the 400 K has a higher

distribution of SGOP values than the 200 K, but based on these simulations, it is not clear why the 600 K should have a smaller distribution of SGOP values, if it is indeed a surrogate for grain size. I don't believe this is addressed. Is this an odd result of the MD simulations? Or a shortcoming of the SGOP value as a surrogate for grain size?

Figure 5a log(Ngrains), it is not clear why if they all started with 250 grains at time zero, why the plot doesn't have all temperatures with 250 grains at time zero? Did coarsening occur during equilibration prior to the measurements in the graph?

Last paragraph of 2.4, "The picture painted in Fig. 5 (b,bottom-right) aligns well with the values in Fig. 5 (b,bottom-left), indicating that our product metric can be used to quantify both the size, shape, and the growth rate of the grains." There is NO evidence that the product metric quantifies anything about shape of the grains.

Second paragraph of 3, "This mapping can be likewise leveraged to output field quantities such as phase order, grain distribution, concentration, stress/strain, and so on." It is not clear to me whether the authors are suggesting that the atomic graph can be used to map field quantities or just the interpolation technique can be used for that since stress/strain and other quantities are already available in per atom values and could be interpolated. I guess I can't find a reference but I believe people have interpolated per atom values before, so I just with the authors to clarify what they are suggesting is new here.

Reviewer #2 (Remarks to the Author):

The manuscript "Quantifying Disorder One Atom at a Time Using an Interpretable Graph Neural Network Paradigm" makes an interesting attempt to produce a particle-level measure of structural disorder, by a somewhat system-agnostic machine-learning method. The specific method use is, I believe, original, and the work seems to be technically sound. However I find the discussion of the physical content of the method and interpretation of the results to be unclear. Furthermore, although the method is novel, the present paper does not appear to produce substantial new scientific results (or if it does, their significance is not clearly explained). Thus, without some comparison to other methods (or a clearer discussion of known faults of existing methods) to make the case that the method itself is a substantial advance, the significance of the results is unclear. I believe the results may be interesting enough to merit publication in Nature Communications, but that substantial revisions are necessary.

I begin with a brief summary of the paper. The authors, in effect, train a neural network to predict the temperature of a sample from the local environment of a particular atom in that sample (although they do not describe it as such). This works insofar as the average prediction matches the true temperature of the sample. They then interpret the single-atom prediction as a quantifier of local disorder—for example, in a high temperature sample, predicting that some specific atoms still have a "low-temperature-like" local environment. The authors prepare a two-phase system and then melt or freeze

one of the phases and observe the changes produced in their order parameter. As a second test case, they observe the evolution of a polycrystalline structure at various temperatures. Finally, they present a simple numerical mapping of their per-particle structural variable to a continuous field.

Definition of SODAS, Validation of SODAS, and Methods

The first area where I see that the manuscript could be improved is its description of the method and its physical meaning. As I understand it, the authors in effect train a graph neural network to predict the temperature from which a local atomic environment was sampled, and interpret the predicted temperature as an indicator of disorder. This ultimate fact, however, is obscured by the layers of their interpretation of what they are doing. My main worry is that a casual reader may not understand that the neural network is effectively predicting temperature and everything relating this to “order” or “disorder” is interpretation (although probably a correct interpretation, in this case). What the authors are doing is perfectly reasonable, but must be more clearly explained. Details:

They begin by alluding to a fictitious temperature T' . The literature is full of different definitions of fictitious temperatures, however when the methods are consulted it is clear that no such concept is really used: the neural network is trained to predict the temperature of a sample. The authors speak of determining T' for a given system using MD simulations but they do not really do any such thing; they set the thermostat to a particular value of $T'(t)$ (since their training simulations heat the system as a function of time). This is either then just the temperature T , or else if the heating rate is too fast and the system is out of equilibrium, it isn't really a precisely defined fictive temperature—just the thermostat temperature, which some degrees of freedom may or may not have equilibrated with. On this particular point, to a reader (such as me) not familiar with the particular MD model of Aluminum studied, it is unclear whether the heating rate is fast or slow, and thus if the system is in equilibrium. It is also unclear why this dynamical protocol, rather than equilibrium samples, is used for training.

The definition of $\gamma(T')$, which serves to re-map this temperature T' to a value between 0 and 1, is also somewhat obscure. It seems like perhaps there is a typo in eq (1), because it is hard to understand exactly how the values are normalized by “ N ” to lie between 0 and 1: For $T=T_d$ the expression on the right is equal to $1/(1+e^{-1}) = 0.73$, and for $T=0$ it is equal to 1. Thus, it is unclear how multiplication by any real number N could change it to range from 0 to 1 rather than 0.73 to 1. The value of “ s ” used in this work also does not appear to be mentioned anywhere.

This definition also raises the problem of the generality of the method. The definition of γ requires two free parameters, a melting temperature and a parameter s which controls the shape of the relationship between T and γ . However, doing so requires us to already know something about the nature of structural order or disorder in a given system. It seems that in e.g. a system where partial ordering sets in gradually, rather than having a well-defined melting temperature, it is not obvious how to proceed, or how one would choose which value of s to use without already knowing what we expect to see.

In a few places the authors interpret γ as saying something about a local contribution to the configurational entropy density. However, this does not appear to be justified—for all we know, it could be e.g. the energy which the neural network is effectively using to distinguish the temperature of

different samples. The authors interpret Fig. 2 as saying something about their MD simulations being “sufficient to capture the configurational entropy present within the material at these temperatures” however the meaning of this statement is unclear and it is unjustified; modulo statements about the average of a function not being the same as the function of the average, the plot, taken at face value, shows that the neural network was successfully trained to predict the temperature of a sample from local environments, and nothing else. The further interpretation should be explained a bit.

The results of Fig S2 is very interesting, and seem to indicate that the method can learn physical behavior which was not put in the definition of γ . However, the relationship between Fig. S2 and Fig 2 is somewhat confusing—what is different in the two plots, such that the deviation around 700K is not present in Fig 2? It is also unclear why using a value of T_d higher than the melting temperature improves sampling of the high-temperature limit; the details provided in the supplemental information do explain the problems caused by this choice but not the reasons for its necessity.

Finally, the discussion around eq (2) is somewhat confusing. The way the authors speak of approximating the function f using a neural network implies that they have in mind a precise, unique definition of the function f which is to be approximated, however the preceding discussion seems to suggest more of a vague conceptual description of such a map (does the value of λ come from the temperature at which a given configuration is most probable? Some precisely defined weighting of the different temperatures at which it may occur? Something else?). It seems to me that the definition of $\lambda(n)$ is perhaps “the value of λ at the temperature the neural network predicts”—is this right? I do not think it needs to be very mathematically precise with some formal guarantees that we are approximating a specifically defined function $f(\{\lambda\}_i)$ (incidentally, what is the set of k ensembles?)—whatever it is, is probably reasonable. But the discussion needs to be clearer.

The discussion of the neural network details is good.

Relevant previous work

A few previous works which use machine learning to try to distinguish ordered and disordered structures should possibly be discussed. <https://pubs.acs.org/doi/10.1021/acsnano.0c07541>, <https://aip.scitation.org/doi/full/10.1063/1.5118867> introduce machine-learned order parameters to distinguish particles belonging to an ordered phase from those belonging to a disordered phase.

<https://aip.scitation.org/doi/10.1063/1.5049849> is also perhaps of some relevance, although they only study the average of their order parameter over the whole system rather than seeking to describe heterogeneities in the system. It is possible that <https://www.nature.com/articles/s41567-020-0842-8> deserves some mention merely for using GNNs to characterize local structure but this is a bit more tenuous. For these second two references I would completely understand if the authors disagreed about their relevance. In general, there is a moderately-sized literature in the glass community on trying to use machine learning to identify local structures with some functional relevance, however likewise I think it is ok to not mention it. The authors briefly touch this literature by citing [24] although I think they may

have misplaced a citation because they cite it as using an atomic “smoothness” metric but the word smoothness does not appear in the cited paper, so perhaps they should double-check this.

The significance of the work

In general there are two routes by which a novel characterization of microstructure can be shown convincingly to be of significance: (1) Its use to learn something novel about the behavior of a material, or (2) some sort of benchmark which convincingly shows that it performs better than, or at least comparably to, other characterizations of microstructure. (To some extent this expectation may vary from community to community within engineering and physics, however.) I think neither of these points has yet been convincingly demonstrated, but that they perhaps could be.

If the results observed in e.g. the grain boundaries are novel, and could not have been obtained using previously existing methods, this would indeed be very exciting, and the authors should explain this more directly.

If, however, the idea is not that the results are novel but that the method is what is novel, then the authors should more clearly demonstrate that their method has some advantage in interpretability (e.g. by clarifying the issues raised earlier in this report), generality and/or performance over existing methods.

The authors try to motivate their approach by arguing that mapping atomic models to continuum ones requires arbitrary approximations. However, it is not clear that any of the various existing parameters used to characterize the degree of order of atomic structure could not be interpolated in exactly the same way as the one that they define. The other advantages which they suggest their approach has (at the bottom of page 2) are likewise not strongly demonstrated: (1) The topology of the network of connected atoms is indeed represented in their calculations as they claim, but it isn't at all clear (given that their system is dense) why this is an important feature, or that it leads to important information being omitted by other methods, (2) given that they are effectively predicting the temperature an atomic neighborhood could have come from, and making the result bounded by passing it through a sigmoidal function, any variable at all could be made bounded by the same procedure, and (3) although the authors speak a couple times of the choice of a GNN as their machine-learning scheme as allowing for “physical interpretability” they never explain why. If we view the procedure as effectively predicting the temperature which is “most typical” for a given local structural environment, then passing that temperature through a nonlinear function to make it bounded between 0 and 1, we naturally wonder why an orientational order parameter, or even the local potential energy (passed through a function to make it bounded between 0 and 1) would have exactly the same properties the authors put forward as advantages of their method.

The authors further state that “Importantly, this procedure leverages a unique feature of SODAS—namely, the ability to provide a physically-motivated and mathematically well-defined threshold for identifying which atoms belong to the grain and which belong to the grain boundary. This process is

extremely challenging using existing methods due to their lack of a continuously and bounded metric.” Again it is unclear why the exact same statements could not equally well apply to a simple structural variable like the local potential energy.

It seems that in any system with a known crystalline order one would be equally well served by using a standard crystalline order parameter. And in any system without a known symmetry, again it seems plausible that e.g. the local potential energy would have similar properties. But it’s quite possible that this is not true, and there is a strong reason I am missing that these simple variables are inadequate. What the manuscript should have is either a precise explanation (with references) of the inadequacies of other methods and how they do not apply to the present method, or (preferably) a comparison of SODAS to a simpler method in the example simulations. A demonstration that in some way such a naïve variable misses something which is captured by SODAS would go a long way toward demonstrating the advantages of the method.

Superiority over other methods is not necessarily a requirement for publication, but I think a clearer demonstration of what is different between SODAS and more naïve characterizations of local disorder is a necessity—as it stands, a reader has no reason to know when this variable would or would not be worth an investment of their time over any other existing method of structural characterization.

Minor issues

The color bars of Fig 3 and Fig 4 are not correct; they appear to be all red.

Typos: Fig.3 caption “reference din” -> “referenced in”, Page 5 “continuously” -> “continuous”

The description of SGOP is somewhat vague. Of course, I understand that it is probably somewhat complicated, and for details it is reasonable to expect the readers to consult the cited paper. However, it is not clear from the description what my lowest-order intuition for what the number means should be—it is described as representing both the grain’s shape and size, but that doesn’t help me interpret a number. In the discussion of figure 5 it is often described as being like the size of a grain—is it approximately the number of atoms in a grain? The authors state in the methods section that it can distinguish even grains with the same number of atoms, however. So in the end I am left uncertain of how to interpret this number. It is also unclear what the product of the average and standard deviation $\mu \sigma$, in Fig 5, means.

The method of interpolation in sec. 2.5 is also not precisely specified—is it linear, or is e.g. a cubic spline used? Does this affect the results?

Finally, it seems like there must be an error in the caption of Fig. 5. It refers to multiple rows, with yellow or green color indicating the temperature of the simulation. However I only see a single row, and no yellow or green.

We thank the reviewers for their critical feedback. We have modified our manuscript based on the recommendations and comments made by the reviewers. To summarize, we have retrained our GNN model of AI using a more robust training data generation scheme. We have also added/redone in a plethora of tests to emphasize how SODAS can be used to better understand the boundary between order/disorder. We have also added a number of comparisons to other commonly used methodologies to showcase how SODAS does indeed outperform them. We hope these additions have addressed the reviewer's concerns and that our manuscript is now ready to be accepted to Nature Communications.

Sincerely,
The Authors

Reviewer #1 (Remarks to the Author):

NCOMMS-22-22606: Quantifying Disorder One Atom at a Time Using an Interpretable Graph Neural Network Paradigm

This paper examines a new structural metric called SODAS and uses it to identify the effective temperature of atoms in a solid.

(a) I write a strong review here with many criticisms. I wish to clarify that I think the metric and even more so, the use of graph neural networks, is a clever and innovative idea that could be useful for machine learning. I think there is a lot of potential in this work. However, as written, it falls short of expectations for Nature Communications. Key among the criticisms is that insufficient information is provided to make the work repeatable; I strongly encourage the authors to make code available or at least have a statement on code availability. Second, many claims about how the method is superior to other methods are given without providing evidence to justify those statements; many are noted in the comments below. In particular, if the authors want to show how much better it is, they could actually calculate the other metrics that are supposedly inferior so they can be contrasted.

We thank the reviewer for these comments, as they are not only fair criticisms but they did highlight a hole in our work. To this end we have expanded sections where we discuss how our models were trained and tested. We have also added in numerous comparisons to commonly used methods to highlight how SODAS outperforms these tools. We note, however, that more complex methods such as ML models, which are not readily available for use, are not considered in this work for comparison to SODAS, as they would require a more-than-significant investment to train/employ. The comments below provide more details of the changes we made.

(b) The literature review is sufficient but I do note one recent paper that gives a bit more perspective on the diverse nature of structural metrics that have been developed (<https://doi.org/10.1021/acs.chemrev.1c00021>); the authors are encouraged to consider how they might redefine their three categories based on this work. There's also some work that considers deviations from solid structures with a local distortion factor that the authors are encouraged to consider how their work is unique since it is also a continuum value (doi: 10.1038/s41467-020-15619-9). Also, other work has used GNNs before too (<https://www.nature.com/articles/s41524-021-00650-1>).

We thank the reviewer for these additions, and we have added these to the manuscript where applicable. We have also added several new references to expand upon previous works. These new references are now listed as [25,26,31,32,35-41].

We would note that for the case of the Distortion factor [32], which the reviewer has provided us a reference with, is built upon the SOAP descriptors, which have been shown to provide non-unique values under broken symmetry conditions. Therefore, while [32] is continuous, it will suffer from the same drawbacks that SOAP suffers from. Also, it requires a non-intuitive process of going from SOAP-to-LDF-to-TDF, wherein numerous levels of noise can creep into the final TDF value. We would argue that SODAS provides an inherently less noisy and more intuitive optimization process.

(c) Bottom of page 2 (Would be great to include line numbers next time), “facilitating universal and intuitive applicability across all atomic structures of a material”. Note that this statement could be deceiving because it can only consider two limiting structures, though perhaps the authors are stating that it can contrast between the two limiting structures regardless of the structure being analyzed. However, in a material that exhibits solid solid phase transformations, would one of these be considered to be more like a liquid than another? Consider these thoughts and how that statement might be revised to accurately portray the potential of the method.

This is an interesting point and one that we did consider but have not explored. The idea of using SODAS to examine solid-solid crystalline transformations would likely not work within the temperature-based framework we have derived, and therefore we have altered the language in our manuscript to be more clear that the current implementation of SODAS works only for order-disorder transformations that can be readily gauged as a function of temperature, such as melting and amorphization.

“(2) SODAS is well-defined and bounded, facilitating universal and intuitive applicability across all order-to-disorder transitions within a material...”

However, we would like to note that the GNN-based framework should have no issue discerning solid-solid crystalline structural transformations, but an alternate route for labelling the data that is not temperature-dependent would have to be taken. In this scenario, the GNN could have multiple outputs rather than a single scalar, perhaps an FCC, BCC, HCP, and disorder parameter, all outputted as a vector. Here, one could not only identify unique geometric environments, but also their corresponding level of disorder. This idea, while interesting, is not explored in this work though.

(d) End of first paragraph of 2.2, “On average, structures between these limits yield intermediate values of λ , as expected.” I find this statement a bit misleading since 1200 K clearly is not as expected. The authors note this later, but they have supplemental Fig 2 that could have easily been incorporated in Figure 2 that would indicate the challenge of SODAS metric in approaching the “liquid” structural arrangements.

We note that in the new Figure 2 that no such problem occurs due to our updated training data generation scheme. We believe the problematic regions in the original manuscript were due to poor sampling of phase space (not enough configurations) within the higher T regions.

(e) Nothing is stated anywhere about whether the simulations have been equilibrated and how that equilibration process occurs. This is particularly relevant for the Solid-Liquid interface simulations. The authors note in the methods (a detail that should be noted in results text) that the liquid region is at 4000K. How do the authors maintain pressure constant when the two parts are at such different

temperatures? Does that lead to negative pressures in the solid region and positive pressures in the liquid region? How does the equilibration (temp and pressure) look when the shared thermostat is then employed? On the one hand it shouldn't be surprising that at 200 K all solidifies or at 1500 K all liquifies, but how was equilibrium obtained prior to these changes? While these don't affect the SODAS analysis, these are odd simulations and it is not clear how typical equilibrium conditions in these and the other simulations have been achieved.

We have updated the Methods section to include a more detailed depiction of the parameters and simulation setup of such simulations. For clarity, the solid-liquid simulation the reviewer references is setup as such: (1) Two regions of the simulation box are identified and isolated from one another thermally, (2) two separate thermostats drive the two regions until equilibrium is reached independently, (3) the regions are combined and the thermostats combined into a single thermostat set at a specific temperature, (4) the system then comes to equilibrium under the new thermostat and the final structures at each temperature are analyzed. We note also that all MD simulations here are performed under NVT conditions.

The author is likely correct that this approach could change the effective pressure of each region, which is a byproduct of using NVT simulations. Although we could track these quantities, we wish to emphasize that the simulations themselves are merely illustrative and not intended to perfectly capture physical constraints. In particular, as the reviewer notes, this choice doesn't significantly affect the SODAS analysis and can still showcase how SODAS can interpret the solid-liquid interface boundary. In fact, we hope that this unusual simulation setup will lend further credence to the broad potential applicability of SODAS.

(f) First paragraph of 2.3, "Bond- angle methods such as CNA and AJ, and more complete methods such as SOAP, often fail to distinguish between perturbed crystalline and disordered atomic environments." The authors offer absolutely no evidence, either by reference or data in their paper, to support this statement. This should be removed unless the authors can provide supporting evidence. Furthermore, while the authors show they can identify the boundary regions with this method, they have not noted anything that shows it to be superior to the other methods mentioned.

We agree that the initial evidence for this was scant, and we have therefore significantly expanded our analysis by adding comparisons to SODAS using 4 commonly used methods for nearly all tests considered in this work. We have also added two references [47,48] from the literature that highlight our point more clearly. Based on these new analyses, we hope that the reviewer will come to the same conclusion that we have in that SODAS generally outperforms these methods while also providing access to information that the other methods cannot quantify. The paper is significantly strengthened by these additions, and we thank the reviewer for encouraging us in this direction.

(g) I have some difficulties with any discussion noting that this method can identify environments. Certainly it is capable of doing that, but because of how it is trained, (from 4.1), "During training, all atoms at a given temperature are assigned the same value of γ . In this way, the value of γ is not directly connected to an atom's local structural geometry. However, at each temperature local atomic geometries bounce around an equilibrium point, which in this case can be thought of as the OK structure. As the temperature increases the magnitude of the displacement from this structure increases, though some atomic neighborhoods may resemble low temperature structural motifs, even at higher overall temperatures." The training is based on temperature, not structures. So, when an atom is assigned a SODAS value, it is its relevance to temperature or common structures at that temperature.

The authors note that it could have that same structure at another temperature so the SODAS value will be different. Furthermore, lots of environments are all being assigned the same temperature. Don't get me wrong, I like the idea of this descriptor, but I think the authors need to be clear about what can actually be distinguished and what can't. What it is trained to capture is temperature, via the structures at each temperature, but thus it is finding lots of structures with the same temperature (and it does this well), but that means the SODAS can tell that the structures of a given temperature exist, not distinguish between different structures, like CNA, SOAP, etc. can do.

It is true that in principle what SODAS does is understand the likely temperature of a given structural motif. In this regard, SODAS effectively predicts temperature. However, SODAS is doing more than simply predicting temperature; it is predicting the most likely value from an ensemble of gamma values that are connected to temperature but are not themselves temperature. This distinction is important as it means SODAS is capturing more than just the temperature of the system; it is capturing how the temperature lands on the order-to-disorder spectrum, and by extension, how a given structural motif exists at some point along the transition between order and disorder. While we agree that the distinction is subtle, and perhaps not explained adequately in the original manuscript, we feel that the difference is important. We further note that our definition of gamma is semi-empirical and may not represent the true, system-dependent physics. This is why the tuning parameter exists in equation 1, to help facilitate the tuning of our definition of gamma for each specific materials system.

We also note that SODAS actually can discern differences between different structures in the same way that SOAP, etc. can; it just doesn't encode this information as a specific model output. Each local structural motif is represented and encoded as a graph that does in fact represent quantitative differences between these local geometries. Ultimately these graphs are fed into the GNN, which encodes them onto a manifold and maps them to a SODAS value. In this regard, the GNN manifold is characterizing the structures the same way SOAP does, but maps this representation to lambda instead of an structure-type identifier. In fact, quantifying differences between these motifs (much as methods such as SOAP can) is actually necessary to know at which SODAS values specific motifs lie. The quantification of local geometry is done indirectly through the graph->GNN->SODAS mapping, but it is still present. The SODAS value is therefore not quantifying absolute differences in local geometries between local motifs, but rather is saying that specific motifs exist at specific points along the order-to-disorder transition.

We note that one could also simply tell the GNN what FCC, HCP, BCC, etc. are and have the model output those values, as hinted in the response to comment (c) above. However, because we are instead interested in quantifying disorder according to the ensemble of local structures, we choose not to output that information.

We have added language to section 2.1 to help clarify these points to the reader:

"It is important to understand that we are arguing a local atomic environment may be represented as the set of points along the order-to-disorder spectrum, rather than its geometric symmetries (or lack thereof). We would argue that this definition provides a more grounded and intuitive representation of the local environment as it can be mapped back onto a physical system, rather than an unsupervised feature vector. While the function f is unknown, it can be approximated. In this work we use a graph neural network scheme to facilitate this approximation, while retaining physical interpretability. It is also important to understand that the GNN is not predicting the temperature of a local environment but rather the point along the order-to-disorder spectrum that the local environment is most likely to exist

at. While this translates, qualitatively, to temperature, it is not explicitly temperature. Fig. 1 outlines the key steps in this process and is discussed in further detail in the methods section.”

(h) Colorbar in Figures 3 and 4 is all red, doesn't go red to blue as suggested it should by the atoms. Also the references in the text to Figure 5a and b are reversed and should be corrected. Did the figure 6 colorbar lambda values get switched? In all the other graphs, lambda=1 was the crystalline region.

We have redone all figures (except for figure 1) in this revision, fixing all issues raised by the reviewer.

(i) While the authors have included many of the necessary details for someone to repeat their work, important details are missing that would make the work very hard to repeat. First off is the missing description for equilibration of the MD simulations. Second, the authors suggest multiple regression layers in Figure 1 but don't ever say exactly how many. I don't think the authors are trying to be intentionally vague, but there is a lot of detail missing or it is hard to follow since it is an involved process. I think the authors must either describe the exact process in more detail, perhaps in supplemental if necessary, or they must provide a script as supplemental material so others can reproduce even a small part of their work. Both are recommended since if they provide a script, people are more likely to apply their methods in the first place.

We thank the reviewer for this comment. We have added additional details in the manuscript for the reproducibility of our work. We have also made plans for the public release of our code.

(j) Additionally, it should be clarified that because of feature sizes, the trained model for SODAS calculations can only be applied to systems with the same number of atoms, right?

SODAS is size-independent and can be applied to systems with any number of atoms. In fact, the model is trained on systems containing only a few thousand atoms but is tested on systems containing millions of atoms.

(k) Another example of missing details is the conversion to an atomic graph mentioned in the second paragraph of 2.4; while the method is published elsewhere, it is not clear how that atomic graph is different from the GNN of this work. Furthermore, the reference paper [38] is also missing lots of details so the reference doesn't appear to be sufficient for reproducibility.

We have removed the graph characterization of microstructures from the manuscript. While we believe that it works, and is very interesting, it appears to have taken away from the primary purpose of that section, which is to highlight SODAS' classification ability, not the graph characterization scheme.

(l) Second paragraph of 2.4, "This process is extremely challenging using existing methods due to their lack of a continuously and bounded metric." This is another unfounded statement, at least with no reference given or evidence shown in the paper to support it. I will also note that there is work in this area that has been incorporated into OVITO for grain segmentation (<https://doi.org/10.1088/0965-0393/24/5/055007>), so while the SODAS method is impressive, it's not the only tool out there.

This is a fair point. We have rewritten our statement to not rely on the notion of "continuous" being the primary motivating factor here, but instead that SODAS is more interpretable. In this way, even the reference of polyhedral template matching (PTM) relies on atoms being too far away from the known

6polyhedral template to identify grain-to-grain boundary transitions. While the final numerical value here is not discrete, its final classifier is. The SODAS classifier itself is continuous, implying that SODAS does not rely on templating to characterize local environments in a binary manner, such as FCC or BCC, etc., but instead gives a continuous value between two binary classifiers. This makes SODAS more powerful than other methods that rely on binary classification because not all environments that are, say, “FCC-like” are the same.

We also note that we have completely rewritten section 2.4 (now 2.5) and make predictions using not only SODAS but also 4 other methods, including the PTM method that is incorporated into OVITO. We hope that these new analyses better emphasize the relative strengths of SODAS, particularly for the problems that we focus on in the manuscript.

(m) Third paragraph of 2.4, “As a result, the distribution of SGOP values provides a physically-intuitive and robust measure of the microstructure. This featurization is a more accurate and unique representation compared to the number of atoms in a grain, the grain radius, or the grain density.” This is another unfounded statement. Grain size and grain radius are both very physically intuitive and I am not even sure what the value of SGOP means. Perhaps it is a robust measure of microstructure but it’s not clear to me what it is a measure of whereas I know exactly what grain radius is.

We have removed the graph characterization of microstructures from the manuscript.

(n) Fourth paragraph of 2.4, “[Figure 5 shows] the interplay between temperature and grain size.” The plot does not show grain size, it shows θ_G , which I can only assume is SGOP. SGOP may be a surrogate for grain size, but it is not grain size.

We have removed the graph characterization of microstructures from the manuscript.

(o) Fifth paragraph of 2.4, “However, a plateau is reached within 2 ns, and no further increase in the mean can be seen.” Figure 5a $\log(\mu)$ has noise that is about the order of any increase or decrease. So, the authors either need to recognize this noise and say any changes are within this noise, or the changes or not noise and each change up or down has meaning. You can’t have both and therefore the quoted statement needs to be adjusted for accurate description of the data. This goes for other comments about data in these graphs of Figure 5.

We have removed the graph characterization of microstructures from the manuscript. In the new Figure 7, we use a combination of 4 values: number of grains, average number of atoms in a grain, number of grain boundary atoms, and the number of mis-classified grain boundary atoms, to quantify each method’s ability to capture the evolution of the polycrystalline systems.

(p) Figure 5 has a product of mean and standard deviation to “gauge both the rate of grain growth and the magnitude of the grains themselves.” Is this an accurate statement? Is this a common statistical value used to capture those kinds of changes? Or is this a creation of the authors and if so, how do they suggest that mean multiplied by standard deviation should capture both rate of growth and the magnitude of the grains themselves since it has units of SGOP^2 ?

We have removed the graph characterization of microstructures from the manuscript.

(q) Figure 5b grain coarsening of the 5 grain structure. It seems obvious that the 400 K has a higher

distribution of SGOP values than the 200 K, but based on these simulations, it is not clear why the 600 K should have a smaller distribution of SGOP values, if it is indeed a surrogate for grain size. I don't believe this is addressed. Is this an odd result of the MD simulations? Or a shortcoming of the SGOP value as a surrogate for grain size?

We have removed the graph characterization of microstructures from the manuscript.

(r) Figure 5a log(Ngrains), it is not clear why if they all started with 250 grains at time zero, why the plot doesn't have all temperatures with 250 grains at time zero? Did coarsening occur during equilibration prior to the measurements in the graph?

We have updated the figure (now fig. 7) to include the $t = 0$ points.

(s) Last paragraph of 2.4, "The picture painted in Fig. 5 (b, bottom-right) aligns well with the values in Fig. 5 (b, bottom-left), indicating that our product metric can be used to quantify both the size, shape, and the growth rate of the grains." There is NO evidence that the product metric quantifies anything about shape of the grains.

We have removed the graph characterization of microstructures from the manuscript.

(t) Second paragraph of 3, "This mapping can be likewise leveraged to output field quantities such as phase order, grain distribution, concentration, stress/strain, and so on." It is not clear to me whether the authors are suggesting that the atomic graph can be used to map field quantities or just the interpolation technique can be used for that since stress/strain and other quantities are already available in per atom values and could be interpolated. I guess I can't find a reference but I believe people have interpolated per atom values before, so I just with the authors to clarify what they are suggesting is new here.

While we agree that interpolating per-atom quantities is not new, we are arguing that because SODAS provides a more physically-interpretable quantity than existing metrics, the interpolation is more accurate. While we only show this for the case of the idealized grain boundaries (Fig. 3), one can imagine applying to every test case considered in this work. Based on the performance of the various methods used here, this point should be clear even though we do not explicitly show it in this work.

Reviewer #2 (Remarks to the Author):

(a) The manuscript "Quantifying Disorder One Atom at a Time Using an Interpretable Graph Neural Network Paradigm" makes an interesting attempt to produce a particle-level measure of structural disorder, by a somewhat system-agnostic machine-learning method. The specific method use is, I believe, original, and the work seems to be technically sound. However I find the discussion of the physical content of the method and interpretation of the results to be unclear. Furthermore, although the method is novel, the present paper does not appear to produce substantial new scientific results (or if it does, their significance is not clearly explained). Thus, without some comparison to other methods (or a clearer discussion of known faults of existing methods) to make the case that the method itself is a substantial advance, the significance of the results is unclear. I believe the results may be interesting enough to merit publication in Nature Communications, but that substantial revisions are necessary.

We thank the reviewer for their comments. We summarize here that we have redone all tests in the work, added a new test (dynamic fracture evolution), and compare SODAS to 4 commonly used methods that are readily available from the community/literature.

We hope that the following details are sufficient to warrant publication.

Definition of SODAS, Validation of SODAS, and Methods

(b) The first area where I see that the manuscript could be improved is its description of the method and its physical meaning. As I understand it, the authors in effect train a graph neural network to predict the temperature from which a local atomic environment was sampled, and interpret the predicted temperature as an indicator of disorder. This ultimate fact, however, is obscured by the layers of their interpretation of what they are doing. My main worry is that a casual reader may not understand that the neural network is effectively predicting temperature and everything relating this to “order” or “disorder” is interpretation (although probably a correct interpretation, in this case). What the authors are doing is perfectly reasonable, but must be more clearly explained. Details:

We have added additional text to section 2.1 to help clarify the complete of the SODAS process to the reader:

“It is important to understand that we are arguing a local atomic environment may be represents as the set of points along the order-to-disorder spectrum, rather than its geometric symmetries (or lack thereof). We would argue that this definition provides a more grounded and intuitive representation of the local environment as it can be mapped back onto a physical system, rather than an unsupervised feature vector. While the function f is unknown, it can be approximated. In this work we use a graph neural network scheme to facilitate this approximation, while retaining physical interpretability. It is also important to understand that the GNN is not predicting the temperature of a local environment but rather the point along the order-to-disorder spectrum that the local environment is most likely to exist at. While this translates, qualitatively, to temperature, it is not explicitly temperature. Fig. 1 outlines the key steps in this process and is discussed in further detail in the methods section.”

We refer the reviewer to our response to comment (g) for reviewer 1. Importantly, while there is a relationships between temperature and disorder, as one would expect, they are not the same. If they were the same, Fig. 2 would indicate a linear relationship between order and temperature, which is not the case.

(c) They begin by alluding to a fictitious temperature T' . The literature is full of different definitions of fictitious temperatures, however when the methods are consulted it is clear that no such concept is really used: the neural network is trained to predict the temperature of a sample.

This a subtle distinction: in reality, the neural network is not predicting the temperature of the sample, it is predicting the most likely gamma value of the local structural motif. Gamma projects the temperature onto a function that attempts to determine where in the abstract manifold space does this temperature lie. This manifold is an abstract quantity that represents the true transition space from solid to liquid and is unknown to us. While we note that gamma is semi-empirical, our validation technique of the average gamma values within a structure shows that our choice of gamma’s functional form is reliable.

(d) The authors speak of determining T' for a given system using MD simulations but they do not really do any such thing; they set the thermostat to a particular value of $T'(t)$ (since their training simulations heat the system as a function of time). This is either then just the temperature T , or else if the heating rate is too fast and the system is out of equilibrium, it isn't really a precisely defined fictive temperature—just the thermostat temperature, which some degrees of freedom may or may not have equilibrated with.

This is correct for the case of training the model. We are using the thermostat temperature as a proxy for our fictitious temperature during training. We argue that is enough of the configuration space is captured (Ergodic constraints), then the mapping between thermostat T and fictitious T is reliable. Again though, for emphasis, the GNN is being trained to predict γ , not the thermostat temperature. We have also made this more clear in the text (see previous comment).

(e) On this particular point, to a reader (such as me) not familiar with the particular MD model of Aluminum studied, it is unclear whether the heating rate is fast or slow, and thus if the system is in equilibrium. It is also unclear why this dynamical protocol, rather than equilibrium samples, is used for training.

We thank the reviewer for this comment. We have generated a new set of training and validation data using only temperature-equilibrated configurations. The new details can be found in section 4.1.

(f) The definition of $\gamma(T')$, which serves to re-map this temperature T' to a value between 0 and 1, is also somewhat obscure. It seems like perhaps there is a typo in eq (1), because it is hard to understand exactly how the values are normalized by “ N ” to lie between 0 and 1: For $T=T_d$ the expression on the right is equal to $1/(1+e^{-1}) = 0.73$, and for $T=0$ it is equal to 1. Thus, it is unclear how multiplication by any real number N could change it to range from 0 to 1 rather than 0.73 to 1. The value of “ s ” used in this work also does not appear to be mentioned anywhere.

One can normalize any function to exist between 0 and 1; if you could not then most machine learning schemes would not work as this is a common practice to make the model more well-behaved during training. The value “ s ” is defined through lines 83-87. We have added our definition of N on lines 84-85.

(g) This definition also raises the problem of the generality of the method. The definition of γ requires two free parameters, a melting temperature and a parameter s which controls the shape of the relationship between T and γ . However, doing so requires us to already know something about the nature of structural order or disorder in a given system. It seems that in e.g. a system where partial ordering sets in gradually, rather than having a well-defined melting temperature, it is not obvious how to proceed, or how one would choose which value of s to use without already knowing what we expect to see.

We thank the reviewer for this comment. While yes, initially one must guess what value of s to choose, there does exist a natural feedback mechanism, which could be automated in practice, for tuning s . Our plot, figure 2, shows the average degree of order (SODAS) for a structure at a given temperature. One can tune s by measuring the deviation of the predicted values by the theoretical γ values. This is because the GNN inherently learns the underlying relationship between γ and temperature. For example, in our original S2, we can see that the predicted SODAS values deviate from the theoretical γ curve at higher temperatures. By tuning s to match the GNN predictions, an easy path exists to automate this process with little-to-no user interference.

We note that in the original S2 the high temperature deviation is also due to poor sampling of our initial training data. Therefore, the GNN's predictions can tell us where our sampling is poor and how to fine-tune s .

Regarding the melting temperature: in fact, the definition of T_d is arbitrary as long as it represents a maximum of disorder for the particular system or problem. In this respect, any temperature can be substituted as the definition of "true disorder". Nevertheless, in our specific case, we note that during the training data generation process it becomes clear which approximate temperature can be used for the melting temperature, as the system either melts or does not melt upon equilibration.

(h) In a few places the authors interpret γ as saying something about a local contribution to the configurational entropy density. However, this does not appear to be justified—for all we know, it could be e.g. the energy which the neural network is effectively using to distinguish the temperature of different samples. The authors interpret Fig. 2 as saying something about their MD simulations being "sufficient to capture the configurational entropy present within the material at these temperatures" however the meaning of this statement is unclear and it is unjustified; modulo statements about the average of a function not being the same as the function of the average, the plot, taken at face value, shows that the neural network was successfully trained to predict the temperature of a sample from local environments, and nothing else. The further interpretation should be explained a bit.

We have rewritten section 2.2, and hence our interpretation of figure 2 has been made more clear. Note that we are not saying that SODAS is configurational entropy but rather that is analogous to it, which we would argue is a valid interpretation. SODAS captures the ensemble of local motifs across the order-to-disorder manifold. We would argue that this is analogous to configurational entropy, which simply captures the entropy contribution from the ensemble of discrete atomic positions. Nevertheless, we agree that the exact mathematical equivalence is unclear and should not have been implied, and the rewrite reflects this sentiment.

(i) The results of Fig S2 is very interesting, and seem to indicate that the method can learn physical behavior which was not put in the definition of γ . However, the relationship between Fig. S2 and Fig 2 is somewhat confusing—what is different in the two plots, such that the deviation around 700K is not present in Fig 2? It is also unclear why using a value of T_d higher than the melting temperature improves sampling of the high-temperature limit; the details provided in the supplemental information do explain the problems caused by this choice but not the reasons for its necessity.

We emphasize again here that the GNN can in fact pick up information that the initial guess of s and T_d may not have had. As stated earlier, this allows the GNN to be iteratively tuned.

Regarding the differences between Fig. 2 and S2, this appears to be due to the sparse sampling in Fig. 2, and was not intended to be deceiving. We simply chose a sparse sampling in Fig. 2 to allow for the inserted 3D images. We have combined S2 and 2 in the updated manuscript for added clarity (new Figure 2).

We also note that the updated training procedure has alleviated any issues with the GNN failing to capture the higher temperature regime.

(j) Finally, the discussion around eq (2) is somewhat confusing. The way the authors speak of

approximating the function f using a neural network implies that they have in mind a precise, unique definition of the function f which is to be approximated, however the preceding discussion seems to suggest more of a vague conceptual description of such a map (does the value of λ come from the temperature at which a given configuration is most probable? Some precisely defined weighting of the different temperatures at which it may occur? Something else?). It seems to me that the definition of $\lambda(n)$ is perhaps “the value of λ at the temperature the neural network predicts”—is this right? I do not think it needs to be very mathematically precise with some formal guarantees that we are approximating a specifically defined function $f(\{\lambda\}_i)$ (incidentally, what is the set of k ensembles?)—whatever it is, is probably reasonable. But the discussion needs to be clearer.

We thank the reviewer for this comment. We are trying to imply that a true mathematical function exists that maps local structural motifs to that abstract transition manifold. While this is not proven, it does make intuitive sense that a true mapping would exist, and this is what we are trying to represent. Note that this is akin to the universal density functional representation within DFT, which has recently become the target of ML models to approximate its form.

However, as stated earlier, what is actually happening here is that a true gamma functional exists for a structural motif, whose complete ensemble is then mapped onto the “true” SODAS value. While we cannot know what the true gamma function is, nor the true mapping from the complete ensemble of gammas to the true SODAS value, we can approximate it with MD-generated samples and an ML model for mapping the discrete ensemble of approximate gamma values to their approximate SODAS value. While this may seem like a stretch mathematically, we believe that this framework makes logical sense, and we leave proving its fundamental nature for future works.

We have attempted to further clarify our description of this process in the text.

Relevant previous work

(k) A few previous works which use machine learning to try to distinguish ordered and disordered structures should possibly be discussed. <https://pubs.acs.org/doi/10.1021/acsnano.0c07541>, <https://aip.scitation.org/doi/full/10.1063/1.5118867> introduce machine-learned order parameters to distinguish particles belonging to an ordered phase from those belonging to a disordered phase.

We thank the reviewer for these additions and they have been added where appropriate [38-39].

In both cases, the ML models are trained using Bond Order Parameters to identify local geometric environments. Therefore, the ML model will ultimately suffer from the drawbacks inherent to those BOPs. We note that we have discussed this in the new section 2.1 from lines 97-101:

“It is important to understand that we are arguing a local atomic environment may be represented as the set of points along the order-to-disorder spectrum, rather than its geometric symmetries (or lack thereof). We would argue that this definition provides a more grounded and intuitive representation of the local environment as it can be mapped back onto a physical system, rather than an unsupervised feature vector.”

(l) <https://aip.scitation.org/doi/10.1063/1.5049849> is also perhaps of some relevance, although they only study the average of their order parameter over the whole system rather than seeking to describe

heterogeneities in the system. It is possible that <https://www.nature.com/articles/s41567-020-0842-8> deserves some mention merely for using GNNs to characterize local structure but this is a bit more tenuous. For these second two references I would completely understand if the authors disagreed about their relevance. In general, there is a moderately-sized literature in the glass community on trying to use machine learning to identify local structures with some functional relevance, however likewise I think it is ok to not mention it. The authors briefly touch this literature by citing [24] although I think they may have misplaced a citation because they cite it as using an atomic “smoothness” metric but the word smoothness does not appear in the cited paper, so perhaps they should double-check this.

We thank the reviewer for their comments [37]. We have corrected the smoothness citation (it was meant to be the distortion factor reference [32]).

The significance of the work

(m) In general there are two routes by which a novel characterization of microstructure can be shown convincingly to be of significance: (1) Its use to learn something novel about the behavior of a material, or (2) some sort of benchmark which convincingly shows that it performs better than, or at least comparably to, other characterizations of microstructure. (To some extent this expectation may vary from community to community within engineering and physics, however.) I think neither of these points has yet been convincingly demonstrated, but that they perhaps could be.

If the results observed in e.g. the grain boundaries are novel, and could not have been obtained using previously existing methods, this would indeed be very exciting, and the authors should explain this more directly.

If, however, the idea is not that the results are novel but that the method is what is novel, then the authors should more clearly demonstrate that their method has some advantage in interpretability (e.g. by clarifying the issues raised earlier in this report), generality and/or performance over existing methods.

We have completely redone sections 2.4 and 2.5 and have introduced section 2.6 to emphasize that we are primarily concerned with showing that SODAS outperforms existing methods. The new figures 4, 6, 7, 8 all showcase the SODAS predictions alongside 4 commonly used methods from the literature (belonging to 2 different classes of characterization scheme). We note that as ML models/codes are not always readily available for the specific systems studied in this work, and training new ones would be too burdensome for this revision, that we restrict ourselves to structural order parameters, and parameterized symmetry functions. We hope these extensive new analyses address the reviewer’s concern and place the research in better context.

(n) The authors try to motivate their approach by arguing that mapping atomic models to continuum ones requires arbitrary approximations. However, it is not clear that any of the various existing parameters used to characterize the degree of order of atomic structure could not be interpolated in exactly the same way as the one that they define. The other advantages which they suggest their approach has (at the bottom of page 2) are likewise not strongly demonstrated: (1) The topology of the network of connected atoms is indeed represented in their calculations as they claim, but it isn’t at all clear (given that their system is dense) why this is an important feature, or that it leads to important information being omitted by other methods,

While we do not validate that the comparison methods in this revision could not be used to perform the atoms-to-continuum mapping shown in Figure 3, we note that these method's shortcomings, as evidenced throughout sections 2.4, 2.5, and 2.6, would ultimately result in mappings that do not yield the same level of fidelity that the SODAS mappings encodes. For example, CAN, AJ, and PTM, both result in a coarse view of the interface regions, likely resulting in information loss when converting from the continuous space to the discrete binary characterization space. For CSP, while it would likely provide more information than the other methods considered, it would still result in a noisy interpretation of the interface region, as evidenced in sections 2.4 and 2.5.

Regarding "(1) The topology of the network of connected atoms", this is not necessarily a unique feature of SODAS, but rather a side-effect of using graph representations rather than local atomic symmetry functions. Graph naturally encode long-range connectivity by nature of their mathematical structure. This information is propagated throughout the GNN during training. Therefore, the GNN inherently learns this information even if it is not explicitly to do so. This is a feature of GNNs that the community believes is responsible for their more well-defined behavior.

(o) (2) given that they are effectively predicting the temperature an atomic neighborhood could have come from, and making the result bounded by passing it through a sigmoidal function, any variable at all could be made bounded by the same procedure,

In principle, the reviewer is correct. However, it is worth emphasizing that within our definition, it is precisely that we are using the ensemble of gamma vales that makes this sigmoidal function have physical meaning. A value of 1 means something different from a value of 0.5 precisely because of how we have defined gamma and lambda. In our view, this physicality—a key strength of our parameterization—would not be transferable to any other random variable that the system inherently possesses.

(p) and (3) although the authors speak a couple times of the choice of a GNN as their machine-learning scheme as allowing for "physical interpretability" they never explain why.

We have added the following text for clarification:

"(3) the compact GNN encoding allows for physical interpretability due to output values being connected to an intuitive property of the system."

We are emphasizing here that it is the fact that the GNN's encoding, which is performed on the graph, which contains all relevant information about the local structure, allows for intuitive understanding of the final output value, which again is connected to the system in an interpretable manner through the gamma-to-lambda mapping.

(q) If we view the procedure as effectively predicting the temperature which is "most typical" for a given local structural environment, then passing that temperature through a nonlinear function to make it bounded between 0 and 1, we naturally wonder why an orientational order parameter, or even the local potential energy (passed through a function to make it bounded between 0 and 1) would have exactly the same properties the authors put forward as advantages of their method.

This is a fair point. In fact, we make no claim that you could not employ exactly the same procedure as SODAS but simply have the GNN output a different property value. In essence, this is exactly what several of the referenced works in the Introduction do. However, the key is to select a disorder property descriptor that is both accurately encoded by the GNN and physically intuitive. We submit that our definition of λ accomplishes this. Still, we acknowledge that other definitions are possible: we could instead have the GNN encode CNA, PTM, CSP, AJ, etc, or even energy as the reviewer suggests, but each of these values has drawbacks. The order parameters all suffer issues with accuracy and interpretability, as evidenced by the new comparisons made in this revision, so by having a GNN predict these values, the GNN will inherit their flaws. As for energy, it could serve as a viable alternative to the set of temperatures used in this work. However, energy is not unique, and different configurations of atoms could have similar local energy profiles; as such, we would argue that our definition of $\gamma \rightarrow \lambda$ is likely to be superior to energy. Nevertheless, a comparison between $\gamma \rightarrow \lambda$ and an energy ensemble (particularly a local energy, as the reviewer suggests) could be an interesting comparison in the future.

(r) The authors further state that “Importantly, this procedure leverages a unique feature of SODAS—namely, the ability to provide a physically-motivated and mathematically well-defined threshold for identifying which atoms belong to the grain and which belong to the grain boundary. This process is extremely challenging using existing methods due to their lack of a continuously and bounded metric.” Again it is unclear why the exact same statements could not equally well apply to a simple structural variable like the local potential energy.

Please see the response to comment (q).

(s) It seems that in any system with a known crystalline order one would be equally well served by using a standard crystalline order parameter. And in any system without a known symmetry, again it seems plausible that e.g. the local potential energy would have similar properties. But it’s quite possible that this is not true, and there is a strong reason I am missing that these simple variables are inadequate.

The reviewer is correct in that systems with order can be mapped onto symmetry functions, and that systems without order could potentially be encoded with something such as local energy (please see response to comment (q) as to why this could potentially be an issue though). However, local symmetry functions alone could have a difficult time distinguishing between degrees of disorder once symmetry is broken, and defining a practical maximum limit of disorder that retains physical meaning becomes difficult. We are also interested in distinguishing homogeneously disordered systems and heterogeneously disordered systems; in this regard, mapping λ onto an ensemble of values (eq. 2) has unique merit beyond the exact definition of λ itself, and the temperature concept is well suited to capturing local structural diversity within an ensemble approach. Although we cannot exclude the possibility that other approaches could be viable, we would argue that SODAS is particularly well suited to the specific problem of disorder.

(t) What the manuscript should have is either a precise explanation (with references) of the inadequacies of other methods and how they do not apply to the present method, or (preferably) a comparison of SODAS to a simpler method in the example simulations. A demonstration that in some way such a naïve variable misses something which is captured by SODAS would go a long way toward demonstrating the advantages of the method.

We agree that with this suggestion. To this end we have added numerous comparisons to existing methods for nearly every test we perform on SODAS, which we feel significantly highlights the advantages of SODAS.

(u) Superiority over other methods is not necessarily a requirement for publication, but I think a clearer demonstration of what is different between SODAS and more naïve characterizations of local disorder is a necessity—as it stands, a reader has no reason to know when this variable would or would not be worth an investment of their time over any other existing method of structural characterization.

We thank the reviewer for this comment. The additions of numerous comparison test cases along with a better examination and description of why SODAS is different/provides access to unique information that existing methods do not provide, are hopefully enough to satisfy this comment.

Minor issues

(v) The color bars of Fig 3 and Fig 4 are not correct; they appear to be all red.

All figures (except for figure 1) have been redone.

(w) Typos: Fig.3 caption “reference din” -> “referenced in”, Page 5 “continuously” -> “continuous”

We have fixed these typos.

(x) The description of SGOP is somewhat vague. Of course, I understand that it is probably somewhat complicated, and for details it is reasonable to expect the readers to consult the cited paper. However, it is not clear from the description what my lowest-order intuition for what the number means should be—it is described as representing both the grain’s shape and size, but that doesn’t help me interpret a number. In the discussion of figure 5 it is often described as being like the size of a grain—is it approximately the number of atoms in a grain? The authors state in the methods section that it can distinguish even grains with the same number of atoms, however. So in the end I am left uncertain of how to interpret this number. It is also unclear what the product of the average and standard deviation $\mu \sigma$, in Fig 5, means.

We have removed SGOP from this manuscript.

(y) The method of interpolation in sec. 2.5 is also not precisely specified—is it linear, or is e.g. a cubic spline used? Does this affect the results?

We have made this more clear in the text. The reviewer is correct though in that the choice of interpolation will affect the results, since different interpolation schemes can result in different coarse-grained values.

(z) Finally, it seems like there must be an error in the caption of Fig. 5. It refers to multiple rows, with yellow or green color indicating the temperature of the simulation. However I only see a single row, and no yellow or green.

All figure captions (except for figure 1) have been redone.

New and updated text:

Lines 38-51

However, quantifying local atomic disorder in a physically motivated way in practice is extraordinarily difficult [19, 20]. Although a number of methods have been proposed to characterize local atomic environments, these methods are often not optimized to magnify the subtle differences present in disordered environments. Existing methods can typically be grouped into three general classes, each of which carries distinct trade-offs: (1) semi-empirical structure factors such as Adaptive Common Neighbor Analysis (CNA) [21], Steinhardt order parameters (SOP) [22], Ackland-Jones order parameters (AJ) [23], atomic excess volume [24], Centrosymmetry parameters (CSP) [25], Scalar Graph Order Parameter (SGOP) [26], and the local atomic environment metric (LAE) [27]; (2) parameterized symmetry functions such as the Smooth Overlap of Atomic Positions (SOAP) [28], Behler-Parinello functions (BP) [29], Moment Tensor Representations (MTR) [30], Polyhedral Template Matching (PTM) [31], Distortion Factors [32], and the Adaptive Generalizable Neighborhood Informed functions (AGNI) [33]; and (3) unsupervised machine learning methods which include graph-based [34–37], order parameter-based [38, 39] and image-based [40, 41] representations.

Lines 66-74

Our approach offers three distinct advantages: (1) the graph representation accurately encodes the topology of the connected network of atoms; (2) SODAS is well-defined and bounded, facilitating universal and intuitive applicability across all order-to-disorder transitions within a material; and (3) the compact GNN encoding allows for physical interpretability due to output values being connected to an intuitive property of the system. The power of this workflow is demonstrated by application to several examples of disordered aluminum systems including solid-solid and solid-liquid interfaces, polycrystalline microstructures, and fracture evolution, and is compared to other methods from the literature such as CNA, AJ, PTM, and CSP.

Lines 98-108

It is important to understand that we are arguing a local atomic environment may be represents as the set of points along the order-to-disorder spectrum, rather than its geometric symmetries (or lack thereof). We would argue that this definition provides a more grounded and intuitive representation of the local environment as it can be mapped back onto a physical system, rather than an unsupervised feature vector. While the function f is unknown, it can be approximated. In this work we use a graph neural network scheme to facilitate this approximation, while retaining physical interpretability. It is also important to understand that the GNN is not predicting the temperature of a local environment but rather the point along the order-to-disorder spectrum that the local environment is most likely to exist at. While this translates, qualitatively, to temperature, it is not explicitly temperature. Fig. 1 outlines the key steps in this process and is discussed in further detail in the methods section

Lines 110-128

We first validate SODAS model by observing where the average SODAS value within several bulk configurations at different temperatures align with the theoretical values of γ , as seen in Fig. 2. Here we see that all atoms in the structure at 0K is uniformly predicted to have $\lambda = 1$, which is indicative of the perfect crystal. In contrast, at 1200K, all atoms indicate λ to be close to 0 due to the structure existing as a melt. On average, structures between these limits yield intermediate values of λ , as expected. In all cases, the average value of λ aligns well with the theoretical value of

y, providing evidence that our methodology, both in conception and implementation, is accurate. A red horizontal line is also drawn in Fig. 2 to provide the reader with a clear understanding of where absolute disorder is represented along the *y* axis.

At the same time, detailed visualization of intermediate-temperature configurations reveals a spectrum of atomic environments covering a range of $\lambda(n)$ in lieu of homogeneously distributed disorder. For example, the second structure in Fig. 2, which represents a structure at roughly 200K, has λ values ranging from 0.5 to 0.95. Accordingly, as described previously, similar atomic environments exist at a range of temperatures but with different degrees of expression according to the average overall level of disorder. Intuitively this makes sense, as the goal of the SODAS metric is to judge the likelihood of an atomic environment existing at an arbitrary point along the thermodynamic spectrum between fully ordered and fully disordered variants. If a unique atomic environment occurs at multiple temperatures, one would expect its λ to be a weighted combination of the individual occurrences of the environment along the temperature spectrum.

Lines 130-133

While the perturbed pristine bulk structure provide a case-study for us to analyze how SODAS performs, most interesting structures contain defects, and varying levels of disorder. To this end, in this section we analyze how SODAS can be used to extract structural information out of solid-solid interface regions.

Lines 162-233

Disordered interface boundaries are notoriously difficult to quantitatively characterize due the inherent complexity and heterogeneity in their local atomic environments. Bond-angle methods such as CSP, CNA, AJ, and PTM often fail to distinguish between perturbed crystalline and disordered atomic environments [50, 51]. These difficulties ultimately make defining interface boundaries challenging. In contrast, the SODAS formalism accomplishes this by providing a continuous metric that allows for a physically justifiable and mathematically rigorous definition of the interface boundary transition.

To this end, we have performed two-phase crystal/liquid CMD simulations at several temperatures (100K, 500K, and 1200K), to observe how several methods classify the unique structural environments present in each scenario. Further details regarding the simulation setup can be found in the Methods section. Figure 4 provides a comparison between SODAS, PTM, and CSP. Here, we examine both the total structure, which again represents the solid-to-liquid transition, as well as a zoom-in view of the interface boundary itself. Figure 4 (a) shows the SODAS characterization of the solid-to-liquid transition, along with a zoomed-in interface boundary region. Here, SODAS correctly identifies the crystalline region as corresponding to structure typically found at low temperatures, as this region is at 100K in Fig. 4 (a). The interface boundary region in the zoomed-in portion also shows a natural gradual progression from solid-to-liquid, which one would expect at equilibrium conditions. There are regions where the interface is more crystalline, and regions that are more disordered, with an intuitive and gradual degradation between these regions. This highlights that SODAS can accurately quantify both the solid and liquid phases, as well as the interface boundary between them.

Figure 4 (b) reference the predictions made by CSP. Here, while CSP does an excellent job of identifying the crystalline region, it's characterization of the liquid phase seems less reliable due to misclassification of various sites throughout the liquid region. This misclassification stems from the fact that CSP works on the notion that values close to zero represent highly ordered crystal structures, while values away from zero represent deviations from those crystal symmetries. While CSP clearly indicates all atoms in the liquid as being away from the corresponding crystal

symmetry, it fails to truly quantify liquid environments from one another. Beyond some threshold, a CSP far away from zero does imply that it is more structurally dissimilar than a value closer to zero but again beyond some threshold. Therefore, it cannot distinguish liquid environments from one another. From Fig. 4 (b) one can also see a more discrete characterization of the interface boundary region, where only a vague guess can be made regarding regions of the boundary that are more "liquid-like" versus "solid-like". As these regions would differ greatly in energy, and therefore properties, one can reason that CSP is not capable of providing an accurate description of this region.

For the cases shown in figure 4 (c-e), which covers AJ, CNA, and PTM, respectively, we examine how the binary classifications schemes perform when characterizing the solid-liquid interface region. In all cases the solid region is well defined, though in both AJ and PTM the liquid region disordered atoms are often classified as a particular solid-phase, indicating a breakdown in the classification algorithm. Within the interface region both AJ and PTM give a seemingly random characterization of structure types. While CNA performs better with a clear mapping between the SODAS and CNA classifications present within the interface, CNA provides a more coarse level of information, giving only the impression of solid-like and liquid-like regions. Overall, SODAS provides a significantly finer and more informative prediction of the solid-liquid interface region.

Figure 5 shows the SODAS characterization of the atomic environments present in the final MD configurations for the three temperatures described earlier. Figure 5 (a) shows the SODAS predictions on the 100K system, with inserted arrows indicating which regions of the structure were initially crystalline and which regions were initially liquid. One can observe, on the initial liquid side, the presence of several disordered regions. These regions are more clearly shown in Fig. 5 (b) due to the specified SODAS range in the colorbar. Intuitively it makes sense that the portion of the structure that was initially a liquid would have defects upon quenching to 100K, while the region that was initially crystalline would not have such defects at 100K.

are not shown here but are implied, with the black dashed line from (a) being present. One can observe, on the initial liquid side, the presence of many disordered regions, with one large disorder patch in the middle. These regions are more clearly shown in Fig. 5 (d) due to the specified SODAS range in the colorbar. Again, this makes sense that the portion of the structure that was initially a liquid would have defects upon quenching to 500K. Importantly, SODAS identifies larger defects present in the initial liquid region than in the initial crystal region, which also makes sense as local environments that lead to larger defects are easier to access kinetically in the liquid phase than they are in the crystal phase. While there are perturbed regions in the initial crystal portion, there are no large-scale defects present.

Figure 5 (e) shows the SODAS predictions on the 1200K system. The inserted arrows from (a) are not shown here but are implied, with the black dashed line from (a) being present. As 1200K is above this interatomic potentials melting temperature, SODAS correctly identifies the total structure as being in the liquid phase. However, we observe patches in the structure that are less disordered than others. These regions are more clearly shown in Fig. 5 (f) due to the specified SODAS range in the colorbar. Interestingly, from Fig. 5 (f), one can see that the regions exhibiting less disorder are more common in the regions that was initially a solid, which makes sense as some level of structural similarity with the solid phase could be present upon melting. It is also possible that the initial crystal region has not yet reached equilibrium with the initial liquid phase. In either case, SODAS captures this trend and allows for a pathway for more complex analysis.

Lines 235-308

The ability to define boundary transitions also enables the identification of larger-scale microstructural features. To demonstrate this capability, we showcase the performance of SODAS when com-

pared to CSP, a-CNA, AJ, and PTM, for classification of both grain boundary and grain regions in dynamic polycrystalline structures. As described in the Methods section, we performed an MD simulation at 600K of a 1.6 million atom FCC aluminum system containing 250 initial grains. Figure 6 provides a visual comparison of the final MD snapshot between the various characterization methods. Here, many grains have coalesced to form larger grains over time as enough kinetic energy was present in the system to overcome large potential energy barriers. Figure 6 (a) provides the SODAS visualization, where one can clearly identify the grain boundary regions (shown in blue), the grains (shown in red), and the transition region between grain and grain boundary (shown in white). Since we are defining grain boundary atoms as atoms having a certain level of disorder, this visualization provides an intuitive thresholding procedure, as one can clearly identify regions based on their level of disorder/order. This prescription will result in some mis-classified atoms, however, as the temperature is increased due to the level of disorder present within the grain regions. The level of mis-classification can be quantified however, and is shown in Fig. 7.

Figure 6 (b-e) provide the visualizations of CSP, PTM, CNA, and AJ, respectively. CSP (b) has difficulty identifying the transition region between grain and grain boundary due to the level of noise present within its characterization. Here, many atoms are mis-classified as grain boundary atoms, and it is clear that there is no intuitive threshold that would allow for precise identification of the two regions. PTM (c) performs better within the grain than CSP, however, it performs worse within the grain boundary region. This is due to the inherent level of disconnectedness found within the boundary regions, where there are significant levels of noise when attempting to discern which atoms belong to the grain boundary. This should be no surprise given PTM's struggles to classify the boundary region of the solid-liquid interface. CNA, shown in Figure 6 (d) provides a better depiction of the GB regions than PTM, though it struggles more within the grains. CNA has difficulty identifying structure types as a level of atomic perturbations, with larger perturbations leading to larger errors in its characterizations of structure types. Therefore, while the boundaries themselves are reasonable, the transition between boundary and grain is not smooth and continuous. Finally, AJ (e) provides the worst characterization of the methods, with thin disorganized boundary regions and chaotic mis-classified grains present throughout the structure.

Figure 7 aims to quantify how both the grain boundaries and grains evolve over time. Figure 7 (a) depicts how the number of grains changes as a function of time for all methods considered in this work. Here, we can see that SODAS was the only method to correctly identify all 250 grains in the initial configuration. As the temperature is increased SODAS predicts a gradual decrease in the number of grains, which coincides with a gradual increase in the number of atoms within the grain regions, as shown in (b). The number of atoms within the grain boundary regions should also decrease in a similar manner, as atoms are leaving the grain boundaries and moving into the grain regions. This is evidenced in (d) where SODAS predicts a similar decrease in the number of grain boundary atoms over time as the growth of grain atoms. (c) shows the number of mis-classified atoms, as described in the Methods section. From (c) we can see that SODAS has the smallest number of mis-classified atom present in the system for all cases except the initial configuration. This is because we've chosen a $\lambda = 0.7$ as our threshold, but at $t = 0$ all atoms in the grain region presumably have a $\lambda = 1$. Due to this, there are additional atoms in the transition region between grain and grain boundary that will get grouped with grain boundary due to their smaller λ value, but actually belong in the grain. While this leads to a larger number of mis-classified atoms in the initial structure, it is important to note that this number is consistent with the remaining structures as the system evolves.

For CSP, Figure 7 (a) indicates a smaller number of grains captured in the initial configuration, followed by an increase in the number of grains to more than the original number. This is due to CSP's noisy characterization, in which many small grains, on the order a few tens of atoms, are

being classified as their own grain instead of belonging to a single larger grain. This leads to a smaller average grain size, as shown in (b). CSP does capture the gentle decrease in the number of grains over time though. Accordingly, CSP is showing a much larger number of grain boundary atoms present in the system, as shown in (d), which we expect due to the noisy characterization. CSP also leads to a larger mis-classification number (c), which aligns with and explains our noisy characterization argument.

For both CNA and PTM, Figure 7 (a) shows a smaller number of initial grains followed by a drastic dropoff as time evolves. This is due to the lack of a true thresholding parameter in the binary classification scheme these methods employ, yielding a more rigid definition of grain and grain boundary atom. While the number of atoms per grain increases in (b), it increases much more sharply than either CSP or SODAS. One can also see a much larger number of mis-classified atoms in (c), again due to issues characterizing perturbed local structures at non-zero temperatures. There is also an increase in the number of grain boundary atoms when compared to SODAS, though the reduction in the grain boundary atoms over time does follow a similar trend.

Finally, for the case of AJ, Figure 7 (a) shows only a single grain present throughout the entire simulation, including the initial structure. This is due to AJ suffering even worse thresholding issues than either CNA or PTM, leading to an unphysical blending of the grain boundary and grain regions. This is also seen in (b) with there being a large number of atoms per grain, as AJ only finds a single grain in the system. This trend aligns with the the number of grain boundary atoms in (d), with AJ registering nearly an order of magnitude fewer grain boundary atoms than any other method. There is also a larger number of mis-classified atoms shown in (c), further solidifying the argument that AJ struggles to concretely define a grain-to-grain boundary transition region, leading to a larger amount of blending between the two domains.

Lines 310-345

Here we examine the performance of various methods at capturing the initiation of tensile fracture. Figure 8 highlights SODAS, CSP, and PTM predictions on snapshots from an MD run of tensile failure. We can see from Fig. 8 (b-e) that SODAS provides a detailed depiction of the atomic environments throughout the entire process. At (b) the cell has undergone minor changes, resulting in some disorder due to the creation of point defects, though most atoms are still very ordered. By (c) enough cell change has occurred to create large disordered regions. Interestingly, there does exist an interface in (c) where small solid regions are surrounded by a disordered medium. In (d) the material has begun to fracture via a void nucleation mechanism. Importantly, SODAS indicates that this void nucleation initiates within one of the disordered regions in (c). Throughout the fracture process, SODAS correctly identifies the disordered regions and maintains a continuous spectrum between the ordered and disordered regions.

Figure 8 (f-i) shows the CSP predictions throughout the fracture process. At (f) the cell has undergone minor changes, though CSP struggles to distinguish the ordered and disordered regions. As stated previously, this is due to a non-intuitive threshold for the transition from ordered to disordered. This ultimately makes it impossible to identify high-level regions within the structure. By (g) enough cell change has occurred to create large disordered regions. While one can qualitatively identify different regions using CSP, the resolution is not enough to quantify them. There is also too much guesswork due to the inherent problem with thresholding in CSP. In (h) the material has begun to fracture via a void nucleation mechanism. However, due to the poor CSP resolution in (g) it is difficult if not impossible to identify what type of local atomic environments were present during the initiation of the void. This implies that CSP would struggle to identify precursor stages

of failure. In (i) CSP correctly identifies the surface regions, and provides a qualitative description of disordered regions within the order regions, though the transition between order and disorder is very abrupt.

Figure 8 (j-m) shows the PTM predictions throughout the fracture process. At (j) the cell has undergone minor changes, and PTM captures these limited changes in atomic structure very well. However, by (k) enough cell change has occurred to create large disordered regions. PTM fails to quantify the order-to-disorder transition in any regard, instead indicating that disordered atoms simply belong to a different crystal structure. This messy characterization provides no clarity with regards to the ground truth. In (l) the material has begin to fracture via a void nucleation mechanism. However, due to the poor PTM resolution in (g) it is difficult if not impossible to identify what type of local atomic environments were present during the initiation of the void. This implies that PTM would also struggle to identify precursor stages of failure. PTM does do a reasonable job in identifying surfaces from bulk, but the abruptness from order-to-disorder is even more stark than CSP. One benefit of PTM, if we assume that it correctly identifies the HCP regions in (l-m), is that it identifies potential dislocation mechanisms within the FCC solid phase, something that neither SODAS nor CSP can do. However, we emphasize here for clarity that methods such as SODAS and CSP are not designed to identify such features. Unlike CSP though, SODAS could be modified to capture such defect systems.\

Lines 379-384

Classical molecular dynamics (CMD), using the LAMMPS software package [53], was used to generate training data for the GNN model. Starting from bulk FCC aluminum (containing 1024 atoms), CMD was performed in the NVT ensemble using the Zhou et al EAM potential [54]. The range of temperatures used for the training data was 50K to 1200K. At each temperature, a NVT simulations was performed for 10 nanoseconds. Data used for training was taken after the 5 ns mark to ensure that only equilibrated configurations were used for training. Further information regarding the training data preparation can be found in the supplemental information

Lines 420-421

All CMD simulations were performed using the LAMMPS software package [53] and the Zhou et al EAM potential [54]

Lines 423-731

2-phase simulations were performed within the NVT ensemble by creating two initial, independent thermostats within a rectangle block of Al containing roughly 51,000 atoms. Both regions are initially crystalline. During the initial stages of the MD simulations, the independent thermostats are used to create a liquid region and a crystal region. Within the liquid region, the thermostat is set to 2000K, which the crystal region is set to 100K. After 5 nanoseconds, allowing for equilibration of the independent phase regions, the two thermostats are removed and replaced by a single new thermostat which acts on the the entire system. This thermostat is set to different temperatures depending on the different scenarios considered. These temperatures are 100K, 500K, and 1200K. The combined system is equilibrated using the single thermostat for 5 nanoseconds.

Lines 433-439

CMD simulations in the NVT ensemble were performed for 4 polycrystalline cases, each with a varying number of initial grains. An initial bulk aluminum system containing roughly 1.6 million atoms was used to construct 6 polycrystalline structures, using the Atomsk software package [62],

containing 250 initial grains. CMD simulations were performed on each case at 600K. NVT simulations were run for approximately 1.5 nanoseconds for each combination of initial structure and temperature. Further details regarding the polycrystalline structures can be found in the Results section as well as the Supplemental Information.

Lines 441-488

Microstructure characterization occurs in 4 stages: (1) calculation of SODAS for all atoms in the system, (2) thresholding of the atomic configuration, based on an atom's SODAS value and subsequent removal of all atoms below the threshold value, (3) conversion of the remaining atoms to a graph representation for the discovery of subgraphs within the graph. Analysis on these subgraphs, such as the calculation of the number of atoms in the subgraph, were then performed. Fig. S3 (b) depicts this workflow visually. While step (1) requires little-to-no input from the user, step (2) requires one to define the level of disorder that needs to be captured when defining the interface regions. This choice highlights the intuitive nature of our proposed methodology, as the threshold value defines the structural properties of the interface region itself, with a near-zero threshold indicating grain boundaries which are extremely disordered and a value close to 1 representing highly crystalline boundaries. In principle, both classes of interfaces can exist within the same structure, which would require a more complex thresholding system, though for this work we assume a uniform local atomic environment amongst all grain boundaries.

For all microstructure characterization tests in this work we employ a thresholding technique to differentiate between atoms belonging to a grain and atoms belonging to a grain boundary. For the case of SODAS we define this threshold 0.7, as this value of λ corresponds roughly to the level of disorder one would expect at 600K. The threshold values at each temperature are defined in a way that minimizes the number of atoms within a grain that may be mistaken as grain boundary atoms. At 600K, which is roughly half of the melting temperature, there is a significant amount of kinetic energy in the system which causes non-trivial levels of atomic perturbation.

The same thresholding technique used for SODAS is also used for CSP. Like SODAS, CSP works on the assumption that non-zero values of CSP represent deviations from a symmetric known crystal structure. However, it is not clear whether or not this trend holds the further one moves from a CSP of zero, implying no obvious cutoff value to distinguish order from disorder. Similar to the SODAS case, we set the 0K CSP threshold at 10^{-5} , and all other CSP thresholds are set at 2, meaning anything CSP value greater than 2 will be considered a grain boundary atom. For PTM, CNA, and AJ, the atomic-level characterization is different than that of CSP and SODAS. Here, environments are encoded as a one-hot vector of known crystal phases. If the stomic environment does not belong to a known reference it is classified as "unknown". Here, any atom classified as "unknown" is determined to be within a grain boundary.

Once atoms have in the system have been thresholded, and all grain boundary atoms have been removed, the remaining system is then mapped onto a graph, G , shown in Fig. S3 (b), where edges are represented by ij pairwise interactions within a $4 \cdot A_{\text{cutoff}}$ radius. A recursive subgraph search algorithm is employed to discover all connected subgraphs, S_G , within the complete graph G . This algorithm is extremely efficient, discovering all subgraphs within a 1.6 million atom system in 1.2 seconds. As all interface atoms were removed prior to the graph construction, all subgraphs in G represent the resulting grains contained within the structure. The total number of subgraphs in the system is equal to the number of grain in the system, and the number of nodes in each subgraph is the number of atoms in a given grain.

We also examine the grain boundary atoms and identify the number of such atoms that have been miss-classified. We define this miss-classification as atoms belonging to the grain boundary designation that do not have a required number of neighbor connections within a cutoff of $3 \cdot A$. Re-

call that here we are only examining grain boundary atoms. Atoms within the true grain boundary should have a large number of grain boundary neighbors, whereas atoms that have been classified as grain boundary atoms but actually lie somewhere within a grain should have a fewer number of grain boundary atoms as neighbors. We use the combination of number of grains detected, atoms per grain distributions, and miss-classified grain boundary atoms to judge a given method's accuracy.

Lines 490-493

A rectangular block of 64,000 atoms was given tensile strain at a constant strain rate of 0.5%/ps, at 100K under NVT conditions. The tensile strain was given in the z direction, and the simulation box was allowed to change along that axis. The box was held constant along the remaining two axis. The simulation was run for 5 nanoseconds.

Lines 495-496

All atomistic visualizations were created using the OVITO software package [63]. Atoms-to-continuum visualizations were done using PyVista [49].

Lines 498-501

For PTM, a root mean square deviation of 0.25 was used. For CNA, an adaptive cutoff radius was employed. For CSP, the number of neighbors was set to 12 since we examined only the FCC phase of Al. We employed the minimum-weight matching convention for CSP. A CSP cutoff of 2 was used as the boundary between order and disorder, due to the location of the first CSP distribution peak.

Figures

Figure 2

Figure 3

Figure 4

Figure 5

Figure 6

Figure 7

Figure 8

REVIEWER COMMENTS

Reviewer #1 (Remarks to the Author):

Thank you for taking the time to address my concerns, it is refreshing after putting much time and effort into providing the review. The revised manuscript is significantly improved. I am also particularly glad to see that the code will be publicly available.

Reviewer #2 (Remarks to the Author):

In our previous reports, the other referee and I had roughly three concerns. (1) We were unsure about some of the data and analyses, particularly the consistency of Figs. 2 & S2, and the SGOP analysis. These concerns were addressed quite well by the revision and need not be discussed further. (2) We disagreed with the authors explanation of their method, feeling that they were ultimately obscuring what was going on and not distinguishing between their interpretation of what their method was achieving and what it was definitely known to be doing. The authors addressed this at least partially with comments to us and revisions, and I will discuss this below. (3) We were unsure of the significance of the work and wanted to see more comparison to existing techniques, or justification of its advantages. The authors addressed this concern with further analyses and changes to the text that I will discuss below.

The other reviewer and I were of the opinion that the authors are essentially training a GNN to predict the temperature a local structure is most likely to have come from, and didn't make this clear to the reader. Firstly, I think that the authors' revisions do make it more obvious to the reader what exactly they are doing, and this is the most important issue, so everything else I have to say is a smaller concern. The authors seem to disagree with us, writing "SODAS is doing more than simply predicting temperature; it is predicting the most likely value from an ensemble of gamma values that are connected to temperature but not themselves temperature". There is I think some sense in which the authors make an important point, and some sense in which they are still missing our point. The training gamma is a monotonic function of temperature. It does not have a precise a priori physically justified definition, but instead two free parameters, which are tuned to find an easier-to-predict function of temperature. The sense in which they may be making an important point is the following: if the structure of the system is not changing much with T in some regions of T, and changes a lot with T in other regions of T, then the value of s and T_d which seem to work best probably have some rough meaning as re-mapping T to a variable which correlates closer-to-linearly with structure than T itself. On the other hand, instead of doing what the authors did, it would have been fundamentally identical to add another set of layers or something at the end of the neural network to learn the inverse mapping from gamma back to T, and say that their neural network is predicting T. This is the sense in which the distinction they make is unimportant. Maybe the particular mapping from T to gamma that works means something. But what it means is ultimately something about which nonlinear functions are more easily learned by the GNN. Does it mean that they are predicting "disorder" rather than "energy" or "density" or any of the other things which correlate with temperature? This is not entirely obvious. All we know

for certain is that SODAS is a purely structural variable which says something about which temperature a structure is most likely to occur at. If it had been trained to predict T^2 , or T^3 , or $1/T$, or $\exp(T)$, that would still be true, and which of those had been easiest to train something might mean something (especially if the model being trained was a simple one, like a linear model rather than a neural network), but since the model being trained can encode nonlinear functions anyway it isn't entirely obvious what it means—and it isn't at all justified that it means “disorder” is being quantified. I do want to emphasize that this isn't a very strong criticism of the results or method, and I think the text they added at least makes it clearer what they did. But I do still think the reader has to do a bit of work to separate out what is interpretation and what is fact.

The authors also often describe their method as “intuitive” and “interpretable” without any real justification of these statements (whose meaning, is indeed, imprecise). I guess they mean in comparison to a purely unsupervised learning method. But I think one should recognize in context that a supervised machine-learned variable is still near the bottom of the ladder of interpretability or physical intuition, below any physical variable with a precise a priori definition. This isn't really an important point, just something for the authors to consider. I don't really know that it means very much to point at figure 3 and say that it is “intuitive”. Why? Compared to what? If the authors have a more precise point in mind, they should say it. When they say “the compact GNN encoding allows for physical interpretability due to output values being connected to an intuitive property of the system” or “While the function f is unknown, it can be approximated. In this work we use a graph neural network scheme to facilitate this approximation, while retaining physical interpretability.”, it doesn't really seem that “retaining physical interpretability” has any meaning at all, other than “we trained a machine learning algorithm to predict a variable which originally had a physical meaning” (and since the mapping from $T \rightarrow \gamma$ is ad hoc, this is only true insofar as temperature had physical meaning). (I am also unsure this latter sentence is strictly correct—the GNN is approximating a mapping from structure to λ and while of course doing so requires it “knowing” that we want the average value of γ to be the same as λ , it doesn't really speak to this distribution. Or maybe it does? I still find the discussion around this function f to be somewhat confusing. In the authors response to my comments they seem to imply the function f maps from structure to λ , but as it is written in the text it seems to map from the list of γ values that this structure has had in some ensemble? I think probably the authors understand this to mean the same thing it is just a bit confusingly explained.) Or when the authors say that choosing a threshold “highlights the intuitive nature of our proposed methodology, as the threshold value defines the structural properties of the interface region itself, with a near-zero threshold indicating...”, I struggle to see how this property wouldn't be shared by any continuous variable that correlates with disorder at all, or why it makes their method particularly “intuitive”.

The sentence “In contrast, the SODAS formalism accomplishes this by providing a continuous metric that allows for a physically justifiable and mathematically rigorous definition of the interface boundary transition.” remains very weakly justified. I do not think any of the other methods discussed by the authors are “physically unjustifiable”, and I don't think there is any justification at all for the claim that this method is somehow more “mathematically rigorous” than any other method. (I also don't think it isn't those things, but it seems that this is implied to be a special property of SODAS.)

The authors also added some comparisons of their method to other structural order parameters to try to show its advantages. These comparisons are definitely appreciated, and I think they are at least partially effective in showing the significance of the work.

In whole, the authors provide a mixture of visual and quantitative comparisons that show SODAS is better behaved for problems of describing heterogeneous systems than other structural order parameters. I think the quantitative comparisons are fairly strong—we see quite clearly that in the case of many grains, where there is a known ground truth (how many grains were seeded in the simulation), the SODAS-based identification of grain boundaries is more accurate in counting the grains than other methods (although do note that if we looked at the orientation of grains using the phase of a crystalline order parameter rather than trying to use a crystalline order parameter to identify grain boundaries, we could presumably do just as well). I am a bit leery of the definition of “misclassified” grain-boundary atoms—the precise number of grain-boundary neighbors required to be considered correctly classified is not specified anywhere in the paper, and it’s not completely obvious that this is a reliable way to identify misclassified atoms, rather than merely reflecting the fact that the other order parameters decided to make narrower grain boundaries (I would be a lot happier if I knew that the number of grain-boundary neighbors you needed to have to be considered correctly classified wasn’t very high.)

Now let’s discuss the visual comparisons to other structural variables. These comparisons are good, but (1) are generally qualitative rather than quantitative, and (2) often lack a clear ground truth, relying on an implicit assumption of what the “right” answer should be (the justification for which isn’t clearly explained) in order to justify the superiority of SODAS to the other structural variables. However I think these are mostly matters of explaining what the comparisons mean—I think they do clearly highlight the differences between SODAS and other methods, which in certain applications should surely be advantages. But maybe they could be explained a bit more clearly.

In Fig 4, we study a liquid-solid interface. I think I would describe these figures as showing that SODAS provides a very smooth field which describes the liquid as homogenous, the solid as homogenous, and describes some smooth variation in the interface between them. If this is what one wants, it’s very nice, and the authors show that to varying degrees the other structural variables lack this behavior. On the other hand, without further evidence of some sort of ground truth it isn’t obvious that, as the authors claim, this makes the other methods wrong. The authors for example refer to identification of solid-like particles in the liquid as “misclassification”. But how are we to know that this doesn’t just reflect genuine structural variation in the liquid, as they discuss favorably about SODAS in Fig. 2 and Fig 5? In some contexts we may wish to resolve the differences between different particles in the liquid (and they may on short timescales at least be genuinely different). Certainly, for certain applications it may be undesirable, however, so again I think the authors basic point is correct, but a bit overstated. Similarly the other structural metrics produce a less “smooth” description of the variation in the plane of the interface, and for some applications this may be undesirable—but the authors seem to be claiming that there is a ground-truth knowledge that this is “wrong”, without showing why. As a particular example, the sentence “As these regions would differ greatly in energy ,and therefore properties, one can reason

that CSP is not capable of providing an accurate description of this region” is not really justified. Of course a discrete classifying variable would remove some information, but the claim that there is some relevant property that is not being captured is not really justified.

The pictures in Figure 6 again show, like in Fig 4, that generally SODAS is “smoother” than the other variables. As mentioned before I wonder if this may confound the comparison of “misclassifications” of grain-boundary atoms.

The simulation in Fig 8 could be a very strong comparison but I wish it were more quantitative. It is a bit difficult to be sure I am understanding the authors’ arguments correctly because the subplot labels appear to be referenced incorrectly in the text and figure caption, but I think I understand correctly. Firstly, a technical point: it would be nice to know which strain/time values each picture represents, and since I am not an expert on Al to know something about the particular strain rate chosen. In particular I wonder if the strain rate may be unreasonably high (well, of course, strain rates in MD simulations are always too high, but more than usual) because it seems surprising that all structural measures basically report a complete disordering of the system before any failure has occurred (b, f, j in the figure, c g k in the caption/text).

If I understand correctly, the authors want to claim that in this same column, the fact that some variation is observed in the SODAS field indicates that it is more predictive of the location of the shear band than the other structural variables. It is impossible to judge this from the figure, however; a quantitative comparison should really be made. (In particular, I don’t doubt that the CSP and PTM will show a weak ability to predict the location of the shear band, I am only unsure from the picture whether or not SODAS is doing any better!) If such a comparison was successful, it would indeed be very nice to see. Some connection to <https://doi.org/10.1103/PhysRevMaterials.4.113609> might be good in this case—note a figure toward the end showing the spatial profile of a structural variable before the shear band has formed.

Overall I think these comparisons between methods are the right idea, but (1) they should be made more quantitative where possible, and (2) the discussion in the text should be clearer about what is “more correct” in reference to some precise ground truth, vs. merely different. (Or if really every difference highlighted by the authors is strictly “more correct”, explain more clearly what prior literature proves this.)

We thank the reviewers for their critical feedback. We have modified our manuscript based on the recommendations and comments made by the reviewers. To summarize, we have added in several new tests/validation of existing tests to (a) be more quantitative where possible and (b) provide external validation of our results where possible. We have also made changes to the language to be clearer with regards to our definitions of terms such as “interpretability” and “order/disorder”. We hope these additions have addressed the reviewer’s concerns and that our manuscript is now ready to be accepted to Nature Communications.

Sincerely,
The Authors

Reviewer #1 (Remarks to the Author):

Thank you for taking the time to address my concerns, it is refreshing after putting much time and effort into providing the review. The revised manuscript is significantly improved. I am also particularly glad to see that the code will be publicly available.

We thank the reviewer for their comments.

Reviewer #2 (Remarks to the Author):

In our previous reports, the other referee and I had roughly three concerns. (1) We were unsure about some of the data and analyses, particularly the consistency of Figs. 2 & S2, and the SGOP analysis. These concerns were addressed quite well by the revision and need not be discussed further. (2) We disagreed with the authors explanation of their method, feeling that they were ultimately obscuring what was going on and not distinguishing between their interpretation of what their method was achieving and what it was definitely known to be doing. The authors addressed this at least partially with comments to us and revisions, and I will discuss this below. (3) We were unsure of the significance of the work and wanted to see more comparison to existing techniques, or justification of its advantages. The authors addressed this concern with further analyses and changes to the text that I will discuss below.

The other reviewer and I were of the opinion that the authors are essentially training a GNN to predict the temperature a local structure is most likely to have come from, and didn’t make this clear to the reader. Firstly, I think that the authors’ revisions do make it more obvious to the reader what exactly they are doing, and this is the most important issue, so everything else I have to say is a smaller concern. The authors seem to disagree with us, writing “SODAS is doing more than simply predicting temperature; it is predicting the most likely value from an ensemble of gamma values that are connected to temperature but not themselves temperature”. There is I think some sense in which the authors make an important point, and some sense in which they are still missing our point. The training gamma is a monotonic function of temperature. It does not have a precise a priori physically justified definition, but instead two free parameters, which are tuned to find an easier-to-predict function of temperature. The sense in which they may be making an important point is the following: if the structure of the system is not changing much with T in some regions of T, and changes a lot with T in other regions of T, then the value of s and T_d which seem to work best probably have some rough meaning as re-mapping T to a variable which correlates closer-to-linearly with structure than T itself. On the other hand, instead of doing what the authors did, it would have been fundamentally identical to add another set of layers or something at the end of the neural network to learn the inverse mapping

from gamma back to T, and say that their neural network is predicting T. This is the sense in which the distinction they make is unimportant. Maybe the particular mapping from T to gamma that works means something. But what it means is ultimately something about which nonlinear functions are more easily learned by the GNN. Does it mean that they are predicting “disorder” rather than “energy” or “density” or any of the other things which correlate with temperature? This is not entirely obvious. All we know for certain is that SODAS is a purely structural variable which says something about which temperature a structure is most likely to occur at. If it had been trained to predict T^2 , or T^3 , or $1/T$, or $\exp(T)$, that would still be true, and which of those had been easiest to train something might mean something (especially if the model being trained was a simple one, like a linear model rather than a neural network), but since the model being trained can encode nonlinear functions anyway it isn’t entirely obvious what it means—and it isn’t at all justified that it means “disorder” is being quantified. I do want to emphasize that this isn’t a very strong criticism of the results or method, and I think the text they added at least makes it clearer what they did. But I do still think the reader has to do a bit of work to separate out what is interpretation and what is fact.

We thank the reviewer for their comment. We think there are 2 points that need clarification, the first is our definition of disorder and the second is the mapping from local atomic geometry to “temperature”. When it comes to disorder, and this partially answers the “intuitive” and “interpretable” comments below, we are defining disorder as the point along a transition from crystal to liquid. We have tried to make this more clear throughout the text. Our definition of disorder is specific to this transition, though it can be extrapolated to any scenario in which one “moves away” from a highly symmetric solid phase. An important point here is that this definition of order/disorder is not easily extendable to scenarios in which one moves from a highly symmetric solid phase to another highly symmetric solid phase. In this scenario, the GNN’s predicted SODAS value, which does indeed depend on temperature in this manuscript, is no longer intuitive and interpretable. Therefore, we have clarified in the text that our definitions and usage of terms such as intuitive and interpretable apply only to scenarios in which one moves from a symmetric structure to a non-symmetric structure. We note that a more general approach to handle an arbitrary A-to-B transition is in the works, though we leave that for a second manuscript as it is itself a large body of work that is separate from solid-to-liquid transitions.

Regarding the mapping from structure-to-temperature, yes, on some level the GNN is predicting temperature, since we are mapping temperature to a non-linear variable called gamma. However, on another level the GNN is not predicting temperature, it is predicting gamma/lambda, with lambda being the pooled value of a set of gamma values for a given structural motif. I think we are both in agreement though that what the GNN is doing, qualitatively, is predicting the most likely temperature that a given structural motif exists at, and this is the purpose of the GNN mapping. In the event that we move from a highly symmetric solid phase to a non-symmetric phase, and the example we used for training is that of a liquid, then we can confidently argue that this structure-to-temperature mapping is indeed a measure of the level of disorder, and even configurational entropy, of the structural motif. This is because our definition of the bounded endpoints is that of an order-to-disorder transition. There are many ways one can approach this problem, and we make no claim that this specific implementation of SODAS is ideal or perfect, but it does work extremely well. We have added language to help the reader understand this this specific version of SODAS is designed for order-to-disorder transitions during training, and one can use the trained model to make predictions for scenarios that one expects the local motifs to follow that trend, that more “ordered” motifs are more like symmetric solid phases, and “disordered” ones are more like non-symmetric phases, as well as added language regarding the future development of a more generalized model.

The authors also often describe their method as “intuitive” and “interpretable” without any real justification of these statements (whose meaning, is indeed, imprecise). I guess they mean in comparison to a purely unsupervised learning method. But I think one should recognize in context that a supervised machine-learned variable is still near the bottom of the ladder of interpretability or physical intuition, below any physical variable with a precise a priori definition. This isn’t really an important point, just something for the authors to consider. I don’t really know that it means very much to point at figure 3 and say that it is “intuitive”. Why? Compared to what? If the authors have a more precise point in mind, they should say it. When they say “the compact GNN encoding allows for physical interpretability due to output values being connected to an intuitive property of the system” or “While the function f is unknown, it can be approximated. In this work we use a graph neural network scheme to facilitate this approximation, while retaining physical interpretability.”, it doesn’t really seem that “retaining physical interpretability” has any meaning at all, other than “we trained a machine learning algorithm to predict a variable which originally had a physical meaning” (and since the mapping from $T \rightarrow \gamma$ is ad hoc, this is only true insofar as temperature had physical meaning). (I am also unsure this latter sentence is strictly correct—the GNN is approximating a mapping from structure to λ and while of course doing so requires it “knowing” that we want the average value of γ to be the same as λ , it doesn’t really speak to this distribution. Or maybe it does? I still find the discussion around this function f to be somewhat confusing. In the authors response to my comments they seem to imply the function f maps from structure to λ , but as it is written in the text it seems to map from the list of γ values that this structure has had in some ensemble? I think probably the authors understand this to mean the same thing it is just a bit confusingly explained.) Or when the authors say that choosing a threshold “highlights the intuitive nature of our proposed methodology, as the threshold value defines the structural properties of the interface region itself, with a near-zero threshold indicating...”, I struggle to see how this property wouldn’t be shared by any continuous variable that correlates with disorder at all, or why it makes their method particularly “intuitive”.

We thank the reviewer for their comments. I think we are in agreement on all points, but that we have not fully defined what we mean by intuitive, or interpretable, etc. Our response above goes into detail about this as well, but essentially we are not claiming that the GNN itself is interpretable but rather that γ is interpretable, which is true, and hence λ is then interpretable. γ is directly connected to temperature, and in the scenario where temperature is directly connected to the level of disorder in the system then so is γ . γ simply serves as a non-linear mapping from temperature to disorder, at a structural/global level. That being said, the GNN isn’t predicting γ , it is predicting λ , which is a global-to-local mapping. This is our function f . We know that a local motif is likely to exist within some region of the true phase space, think of it as an average point along a transition from phase A to phase B.

This is a true function, it must exist, but we have no idea what this function is and likely cannot know because it would require us to know all possible configurations of atoms between phase A and B. Therefore, we sample the pathway via MD and train the GNN to approximate this function f based on our rough guess at the actual topology of this function. For the scenario of solid-to-liquid, our definition of γ , which again is our guess at the topology, which we are using loosely, works very well. The GNN is approximating f via a high-dimensional parameterized function by using γ as our approximate form we want it to learn, since on average λ should oscillate around some average line through the true phase space, assuming equilibrium conditions. In this way, our prediction of

lambda is interpretable and intuitive because gamma is interpretable and intuitive, because temperature is interpretable and intuitive.

We agree that when temperature fails to be interpretable, such as some A-to-B transition that does not depend on temperature, then this specific setup likely won't work, and a more generalized framework is needed (which we are working on). However, when temperature is interpretable SODAS is interpretable. Why not predict temperature directly? The order-to-disorder spectrum is connected to temperature in a non-linear way, which we believe is more likely to remain interpretable if we give the GNN a function to learn that is interpretable, otherwise the GNN may learn some unintuitive non-linear function that maps local motif to temperature. It may work, but it may not be obvious that 0.5 means "a level of disorder equivalent to what you would see at 600K". This is what we mean by "interpretability", and we have attempted to make that clearer to the reader.

The sentence "In contrast, the SODAS formalism accomplishes this by providing a continuous metric that allows for a physically justifiable and mathematically rigorous definition of the interface boundary transition." remains very weakly justified. I do not think any of the other methods discussed by the authors are "physically unjustifiable", and I don't think there is any justification at all for the claim that this method is somehow more "mathematically rigorous" than any other method. (I also don't think it isn't those things, but it seems that this is implied to be a special property of SODAS.)

We have tempered the language around these assertions.

The authors also added some comparisons of their method to other structural order parameters to try to show its advantages. These comparisons are definitely appreciated, and I think they are at least partially effective in showing the significance of the work.

In whole, the authors provide a mixture of visual and quantitative comparisons that show SODAS is better behaved for problems of describing heterogeneous systems than other structural order parameters. I think the quantitative comparisons are fairly strong—we see quite clearly that in the case of many grains, where there is a known ground truth (how many grains were seeded in the simulation), the SODAS-based identification of grain boundaries is more accurate in counting the grains than other methods (although do note that if we looked at the orientation of grains using the phase of a crystalline order parameter rather than trying to use a crystalline order parameter to identify grain boundaries, we could presumably do just as well).

I am a bit leery of the definition of "misclassified" grain-boundary atoms—the precise number of grain-boundary neighbors required to be considered correctly classified is not specified anywhere in the paper, and it's not completely obvious that this is a reliable way to identify misclassified atoms, rather than merely reflecting the fact that the other order parameters decided to make narrower grain boundaries (I would be a lot happier if I knew that the number of grain-boundary neighbors you needed to have to be considered correctly classified wasn't very high.)

We thank the reviewer for the chance to clarify these details. We did not use a predefined number of neighbors but rather a radial cutoff to generate the neighbor list. We defined our threshold for misclassification as 3 neighbors. Initially, we removed all atoms that we defined as grain atoms (how will depend on the specific method we are examining) and then calculated the number of neighbors within a 3A cutoff, again only using the remaining atoms. If an atom had 4+ neighbors then we have called it correctly classified as a GB atom, and if it had less than 4 (3 or less) then it was counted as a

misclassified GB atom. Admittedly this is not an ideal metric, but it does give us information about situations where a given metric has many atoms within the grain regions that it is labelling the same way as atoms within the GB regions.

Now let's discuss the visual comparisons to other structural variables. These comparisons are good, but (1) are generally qualitative rather than quantitative, and (2) often lack a clear ground truth, relying on an implicit assumption of what the "right" answer should be (the justification for which isn't clearly explained) in order to justify the superiority of SODAS to the other structural variables. However I think these are mostly matters of explaining what the comparisons mean—I think they do clearly highlight the differences between SODAS and other methods, which in certain applications should surely be advantages. But maybe they could be explained a bit more clearly.

In Fig 4, we study a liquid-solid interface. I think I would describe these figures as showing that SODAS provides a very smooth field which describes the liquid as homogenous, the solid as homogenous, and describes some smooth variation in the interface between them. If this is what one wants, it's very nice, and the authors show that to varying degrees the other structural variables lack this behavior. On the other hand, without further evidence of some sort of ground truth it isn't obvious that, as the authors claim, this makes the other methods wrong. The authors for example refer to identification of solid-like particles in the liquid as "misclassification". But how are we to know that this doesn't just reflect genuine structural variation in the liquid, as they discuss favorably about SODAS in Fig. 2 and Fig 5? In some contexts we may wish to resolve the differences between different particles in the liquid (and they may on short timescales at least be genuinely different). Certainly, for certain applications it may be undesirable, however, so again I think the authors basic point is correct, but a bit overstated. Similarly the other structural metrics produce a less "smooth" description of the variation in the plane of the interface, and for some applications this may be undesirable—but the authors seem to be claiming that there is a ground-truth knowledge that this is "wrong", without showing why. As a particular example, the sentence "As these regions would differ greatly in energy, and therefore properties, one can reason that CSP is not capable of providing an accurate description of this region" is not really justified. Of course a discrete classifying variable would remove some information, but the claim that there is some relevant property that is not being captured is not really justified.

We have added a new test, which is included in the SI along with text in the main manuscript, that examines the average order parameter value as a function of the y-axis (the axis in which we move from liquid to solid). We have also added references to both computational and experimental works that show the solid-liquid interface of Al should slowly and continuously transition from solid-to-liquid with a boundary region of around 3-4 atomic layers. The new test, shown in the SI, confirms that SODAS matches this description more accurately than the other methods, with CNA doing a reasonable job as well. This test confirms that SODAS matches experimental evidence regarding the topology of the solid-liquid interface.

The pictures in Figure 6 again show, like in Fig 4, that generally SODAS is "smoother" than the other variables. As mentioned before I wonder if this may confound the comparison of "misclassifications" of grain-boundary atoms.

We have changed any reference to the word smoother with regards to Fig. 6, as that isn't really our goal for the takeaway. What we are trying to say here, and Fig. 6 is a qualitative figure, is that SODAS provides both a reliable way to identify grain boundary atom from grain atoms and provides a more sensitive/finer depiction of where the grain boundary begins and ends. While this is not completely

quantitative, we do note that there exists a plethora of literature to suggest that grain boundaries in Al crystals, especially at higher temperatures, so not exist as 1-2 atom thick boundaries, nor as ideal stacking faults. We have noted this in the main text.

The simulation in Fig 8 could be a very strong comparison but I wish it were more quantitative. It is a bit difficult to be sure I am understanding the authors' arguments correctly because the subplot labels appear to be referenced incorrectly in the text and figure caption, but I think I understand correctly. Firstly, a technical point: it would be nice to know which strain/time values each picture represents, and since I am not an expert on Al to know something about the particular strain rate chosen. In particular I wonder if the strain rate may be unreasonably high (well, of course, strain rates in MD simulations are always too high, but more than usual) because it seems surprising that all structural measures basically report a complete disordering of the system before any failure has occurred (b, f, j in the figure, c g k in the caption/text).

If I understand correctly, the authors want to claim that in this same column, the fact that some variation is observed in the SODAS field indicates that it is more predictive of the location of the shear band than the other structural variables. It is impossible to judge this from the figure, however; a quantitative comparison should really be made. (In particular, I don't doubt that the CSP and PTM will show a weak ability to predict the location of the shear band, I am only unsure from the picture whether or not SODAS is doing any better!) If such a comparison was successful, it would indeed be very nice to see. Some connection to <https://doi.org/10.1103/PhysRevMaterials.4.113609> might be good in this case—note a figure toward the end showing the spatial profile of a structural variable before the shear band has formed.

The strain rate used in these simulations used in this simulation was 0.5%/ps, which is extremely high. However, the purpose of this simulation was not to be correct with regards to reality, but rather to showcase that SODAS can identify regions that are extremely different than what it was trained on (ex: surfaces). The fact that SODAS works as a classifier of something like shear band during an extreme simulation such as this, gives us confidence that SODAS would perform adequately if not better under a more realistic scenario.

We agree with the reviewer that this scenario would be a nice example if it were more quantitative. To that end, we have calculated the shear band location plots for all methods, including our reference method of D^2 , which we will call the ground truth. For this comparison, probability for our characterization methods is defined as the likelihood, for a given x, y, or z-coordinate (note that we show these separately as 3 different plots), we take the average order parameter value at that given x, y, or z-coordinate. Therefore, the Z-coordinate plot will be the most informative, as the average order parameter value along the z-axis will be less affected by averaging than along the x or y axes (since the simulation box is tall but narrow). This plot shows that SODAS performs the best at not only locating the shear band but also predicting where it's next location will be, providing a quantitative measure of SODAS' accuracy for tensile strain/failure simulations. This data has been included in a new figure 8, which replaces the old figure 8, with that figure being placed in the SI.

Overall I think these comparisons between methods are the right idea, but (1) they should be made more quantitative where possible, and (2) the discussion in the text should be clearer about what is "more correct" in reference to some precise ground truth, vs. merely different. (Or if really every difference highlighted by the authors is strictly "more correct", explain more clearly what prior literature proves this.)

We hope that these changes have been satisfactory.

REVIEWER COMMENTS

Reviewer #2 (Remarks to the Author):

My previous review discussed a mixture of concerns about how the method & its physical meaning are explained, and a desire for more quantitative comparisons between the different methods.

On the issues of explanation (the meaning of the disorder parameter γ , the description of the method as "intuitive" and "interpretable") I think we have more or less hit a compromise I am willing to accept. I still would not describe the method the way the authors do, but everything they say is factually correct and I think this is just an acceptable difference of interpretation.

On the technical, characterization-of-SODAs-on-examples side, I think the authors additions are a nice improvement and more or less satisfy me. A few technical comments on figure 8 that could perhaps be clarified in the final version:

1. The explanation in lines 334-342 of what the z axis means for the structural indicators is clear, but it is unclear whether D2 is treated in the same way as CSP/SODAS or something else.
2. The discussion of "predicting shear bands" is a bit confusing. D2 is a measure of how much atoms have already moved (or perhaps are about to move in the next small time step, both have been done before and the authors do not make this completely clear--it might be good to add a word like "cumulative" or "instantaneous" the first time D2 is mentioned), NOT a structural predictor. If D2 has a strong banding structure, that certainly indicates the presence of a shear band. However, that does **not** mean that, as the authors say in lines 352-354, that any method that predicts a shear band from $t=0$ is necessarily wrong. D2 being zero just means that nothing has moved yet. If a structural indicator predicts the future rather than being merely an indicator of the past, this would generally be viewed as a good thing rather than a bad thing.

Similarly in the discussion at 362-369 the authors consider it bad that the other methods "predict three peaks" at $t=3\text{ns}$ but D2 "predicts" (i.e. currently shows) 2 peaks. But at $t=4\text{ns}$ all three peaks are now present---if the other methods predicted their appearance in advance, would that be a bad thing?

Now, it is possible I am misunderstanding the authors here. They might genuinely view D2 as a predictor of the location of the shear band, insofar as "past displacement predicts future displacement" could be considered a reasonable heuristic. However, I would not really consider this to be a reliable ground truth, rather than the actual future location of displacement.

In fact, since the authors proceed all the way to fracture, it isn't even clear a dynamical ground truth "variable" is needed---it would be just as reasonable to discuss how the dynamical indicator profiles at various times line up with the final fracture locations, which are perfectly obvious from pictures of the simulations.

3. That being said, I think it's still true that the peak present in the various other structural indicators at $t=0$ doesn't seem to correctly predict the location of the shear band in the future. And it might be considered a good thing that at $t=2\text{ns}$ SODAS seems to predict the future development of large displacements near the right boundary

Overall I think this quantitative characterization is good, but in the final paper it would be good to revise this discussion a bit.

We thank the reviewers for their critical feedback. We have modified our manuscript based on the recommendations and comments made by the reviewers. We hope these additions have addressed the reviewer's concerns and that our manuscript is now ready to be accepted to Nature Communications.

Sincerely,
The Authors

Reviewer #2 (Remarks to the Author):

My previous review discussed a mixture of concerns about how the method & its physical meaning are explained, and a desire for more quantitative comparisons between the different methods.

On the issues of explanation (the meaning of the disorder parameter γ , the description of the method as "intuitive" and "interpretable") I think we have more or less hit a compromise I am willing to accept. I still would not describe the method the way the authors do, but everything they say is factually correct and I think this is just an acceptable difference of interpretation.

On the technical, characterization-of-SODAs-on-examples side, I think the authors additions are a nice improvement and more or less satisfy me. A few technical comments on figure 8 that could perhaps be clarified in the final version:

1. The explanation in lines 334-342 of what the z axis means for the structural indicators is clear, but it is unclear whether D2 is treated in the same way as CSP/SODAS or something else.

We thank the reviewer for this comment, and we agree that the discussion around this point was unclear. We have modified the text accordingly (lines 343-347). To summarize, we use kernel density estimation to find the most probably regions along the z-axis with values associated with "disorder". This is performed for all methods used in this work, including D2. The word "average" was meant to convey that the KDE peaks will represent areas along the z-axis that exhibit the highest level of average disorder for that z-coordinate, but it is not explicitly the calculation of the average. We have clarified this in the updated text.

2. The discussion of "predicting shear bands" is a bit confusing. D2 is a measure of how much atoms have already moved (or perhaps are about to move in the next small time step, both have been done before and the authors do not make this completely clear--it might be good to add a word like "cumulative" or "instantaneous" the first time D2 is mentioned), NOT a structural predictor. If D2 has a strong banding structure, that certainly indicates the presence of a shear band. However, that does *not* mean that, as the authors say in lines 352-354, that any method that predicts a shear band from $t=0$ is necessarily wrong. D2 being zero just means that nothing has moved yet. If a structural indicator predicts the future rather than being merely an indicator of the past, this would generally be viewed as a good thing rather than a bad thing.

We thank the reviewer for their comment. We have made changes to lines 337-340 to make the discussion around why we are using D2 as a reference method clearer. We also agree that the discussion around what deviations from D2 mean can be made clearer. What we are arguing is that, when a given method predicts peaks in their respective levels of disorder that do not end up agreeing with D2 either in the present or in the future then it is likely that the method in question is mischaracterizing

environments are disordered when they are not. We did not discuss instances where a method like CNA does predict future behavior in D2 and agree that this is a very important piece of missing discussion. We have made this clearer in the text throughout that section.

Similarly in the discussion at 362-369 the authors consider it bad that the other methods "predict three peaks" at $t=3\text{ns}$ but D2 "predicts" (i.e. currently shows) 2 peaks. But at $t=4\text{ns}$ all three peaks are now present---if the other methods predicted their appearance in advance, would that be a bad thing?

Please see the previous comment.

Now, it is possible I am misunderstanding the authors here. They might genuinely view D2 as a predictor of the location of the shear band, insofar as "past displacement predicts future displacement" could be considered a reasonable heuristic. However, I would not really consider this to be a reliable ground truth, rather than the actual future location of displacement.

This is an important point that we have attempted to make clearer in the text. The main idea that we are trying to convey is that, the methods considered in this work should either (1) match the predictions of D2 or (2) predict future behavior in D2. If they do not fall into either of those bins then it is a reasonable conclusion that they are misclassifying environments. Since D2 has been used in the past to identify the approximate locations of shear bands then any method that attempts to predict these locations should fall into either (1) or (2). Methods that do not likely cannot be used as shear band predictors. We have made this clearer in the text throughout lines 350-390.

In fact, since the authors proceed all the way to fracture, it isn't even clear a dynamical ground truth "variable" is needed---it would be just as reasonable to discuss how the dynamical indicator profiles at various times line up with the final fracture locations, which are perfectly obvious from pictures of the simulations.

We have added a small modification to figure 8 to show the region in which fracture initiates. This region does line up well with the primary peak in D2 and SODAS but not the primary peak for all other methods. The agreement between these methods and D2 is better before and after fracture, but breaks down during fracture, indicating that SODAS is a better instantaneous predictor.

3. That being said, I think it's still true that the peak present in the various other structural indicators at $t=0$ doesn't seem to correctly predict the location of the shear band in the future. And it might be considered a good thing that at $t=2\text{ns}$ SODAS seems to predict the future development of large displacements near the right boundary

We have added a short discussion around the observation that AJ and a-CAN seem to predict some future behavior in D2 from $t=0\text{ns}$ to $t=2\text{ns}$, but both CSP and PTM do not show any future predictive behavior. However, AJ and a-CAN do not show the same level of predictive behavior during and after fracture, indicating that there is a limit of geometric complexity in which these methods are no longer sufficient in predicting future behavior or atomic disorder. We have added this discussion in the text between lines 378-84.

Overall I think this quantitative characterization is good, but in the final paper it would be good to revise this discussion a bit.

We thank the reviewer for their comments and hope our additions satisfy their questions/requests.

REVIEWERS' COMMENTS

Reviewer #2 (Remarks to the Author):

I think the updated discussion of Figure 8 describes what we should conclude from the data well. The final picture is that SODAS is not unambiguously "better" than the other methods, but that is fine, in my opinion. In the end the different behaviours of the different structural indicators have been well characterized, and I would be happy with the paper being published in this form.